



# On the Flood Peak Distributions over China

Long Yang[1], Lachun Wang[1], Xiang Li[2], and Jie Gao[3]

[1]School of Geography and Ocean Science, Nanjing University, Nanjing, Jiangsu province, China
[2]China Institute of Water Resources and Hydropower Research, Beijing, China
[3]China Renewable Energy Engineering Institute, Beijing, China

**Correspondence:** Long Yang (yanglong86123@hotmail.com)

**Abstract.** Time series of annual maximum instantaneous peak discharge from 1120 stations with record lengths of at least 50 years are used to examine flood peak distributions across China. Abrupt change rather than slowly varying trend is the dominant mode of the violation of stationary assumption for annual flood peaks over China. The dominance of decreasing trends in annual flood peak series indicates a weakening tendency of flood hazard over China in recent decades. Delayed (advanced) occurrence
of annual flood peaks in southern (northern) China point to a tendency for seasonal clustering of floods across the entire country. We model the upper tails of flood peaks based on the Generalized Extreme Value (GEV) distributions for the stationary series, and evaluate the scale-dependent properties of flood peaks. The relations of GEV parameters and drainage area show spatial contrasts between northern and southern China. Weak dependence of the GEV shape parameter on drainage area highlights the critical role of space-time rainfall organizations in dictating the upper tails of flood peaks. Landfalling tropical cyclones
play an important role in characterizing the upper-tail properties of flood peak distributions especially in northern China and southeastern coast, while the upper tails of flood peaks are dominated by extreme monsoon rainfall in southern China. Severe flood hazards associated with landfalling tropical cyclones are characterized with tropical cyclones experiencing extratropical transition, and persistent moisture transport/interactions with regional topography as demonstrated by Typhoon Nina (1975).

## 1 Introduction

We examine flood peak distributions over China based on 1120 stream gauging stations with continuous records of annual maximum flood peaks for at least 50 years. The ultimate goal of our study is to provide improved understandings on the nature of upper tails of flood peaks and innovative methods for flood frequency analysis in a changing environment. The central themes of this study are (1) stationarity of annual flood peaks (for both peak magnitude and timing), (2) mixture of flood-generation systems, (3) spatial heterogeneity of flood peak distributions, and (4) the impacts of tropical cyclones on the
upper-tail properties of flood peak distributions.

 Hydrological regimes in most river basins over China, like the rest of the world, have experienced strong anthropogenic influences (i.e., river regulations, land use changes). Human-related impacts on flood hydrology are further complicated by detectable changes in external factors that are critical for flood-generation processes, such as temperature and extreme rainfall, even though it remains unsettled whether the changes are due to natural climate variability or human-induced climate change
(e.g., Held and Soden, 2006; Marvel and Bonfils, 2013; Trenberth et al., 2015; Schaller et al., 2016; Risser and Wehner, 2017;





Eden et al., 2017). The staionarity assumption of flood series has been questioned and debated in the scientific community (Milly et al., 2008; Montanari and Koutsoyiannis, 2014; Salas et al., 2018). Extensive studies on the stationarity of annual maximum flood peaks have been carried out in many parts of the world (e.g., Robson et al., 1998; Robson, 2002; Franks and Kuczera, 2002; Villarini et al., 2009; Petrow and Merz, 2009; Villarini et al., 2011; Ishak et al., 2013; Tan and Gan, 2014;

Mediero et al., 2014; Do et al., 2017; Hodgkins et al., 2019). There are also some regional studies across China (e.g., Zhang et al., 2016, 2014, 2018b; Liu et al., 2018). A nation-wide investigation on the stationarity of annual maximum flood peaks over China, however, is still missing, and is the principal focus of our study. The exceptional dataset of flood records in China, as demonstrated in the present study, will provide additional evidence for detectable changes in flood series in a changing environment. In this study, we do not aim to do attribution analysis for the changes in annual flood peaks for each river basin

(like e.g., Hodgkins et al., 2019), but to highlight possible factors that induce the changes. Similar with the study by Villarini and Smith (2010) in the eastern United States, we identify the dominant modes of violation for the stationarity assumption in annual maximum flood peaks over China. Better understanding of historical changes in annual flood peaks is of paramount importance for constraining model-based projections of flood hazards in a changing environment (e.g., Milly et al., 2002; Hirabayashi et al., 2013; Dankers et al., 2014; Arnell and Gosling, 2016).

We examine the upper tails of flood peaks over China based on the statistical framework of the generalized extreme value (GEV) distributions (similarly see e.g., Katz et al., 2002; Morrison and Smith, 2002; Villarini and Smith, 2010; Barros et al., 2014; Bates et al., 2015; Gaume, 2018; Smith et al., 2018). Special focus is placed on the spatial heterogeneity of flood peak distributions over China. Previous studies show strong dependence of location and scale parameters for the GEV distributions on drainage area, while the GEV shape parameters only weakly depend on drainage area (Morrison and Smith, 2002; Villarini

and Smith, 2010). Weak dependence of the GEV shape parameters on drainage area indicate scale-independent properties of the upper tail of flood peak distributions, and highlight additional factors (e.g., space-time rainfall organizations) in determining the upper tails of flood peaks. Yang et al. (2013) identified a spatial contrast of extreme rainfall distributions between northern and southern China and point to contrasting flood hydroclimatology across the country. We propose that similar spatial contrasts also exist in flood peak distributions across China, and highlight the necessity of improved procedures for regional flood

frequency analysis with spatial heterogeneity in flood hydroclimatology considered.

In addition to annual maximum flood peak magnitude, we examine the timing of annual maximum flood peak (as represented by day of the year) in this study. Analysis on the timing of annual flood peaks (e.g., seasonality) can shed light on flood-generating mechanisms, and is an important aspect of flood frequency analysis (Hirschboeck, 1988; Singh et al., 2005; Leonard et al., 2014; Brooks and Day, 2015; Yan et al., 2017, 2019). Annual flood peaks resulted from different flood-generation

mechanisms violate the assumption of homogeneous flood population that most conventional methods for flood frequency analysis rely on. This is, however, often the case for many regions in the world (e.g., Jarrett and Costa, 1988; Villarini, 2016; Blöschl et al., 2017; Smith et al., 2018; England et al., 2018). For instance, flooding in the eastern United States is under mixed controls of extratropical systems and landfalling tropical cyclones, with relative importance varies spatially (Smith et al., 2011). The upper-tail properties of flood peak distributions are frequently determined by extreme events resulted from

anomalous flood agents for particular regions (the notion of "strange floods", see Smith et al., 2018). We expect annual flood





peaks to be characterized with a mixture of flood-generating mechanisms over China, due to the geographic location of China in a monsoon-climate region and on the western margin of the north Pacific basin for tropical cyclones. We characterize the relative importance of monsoon-related systems and tropical cyclones in dictating the upper tails of flood peaks across China. Consistent changes in flood peak timing over Europe indicate a clear signal of climate impact on the seasonality of European

flood records (Blöschl et al., 2017). Villarini (2016) does not find a strong signal of temporal changes in the seasonality of annual maximum discharge over the continental United States. Changes in the seasonality of annual flood peaks across China is still lacking, despite its strong implication for flood hazards and water resources management. In this study, we examine changes in the timing of annual maximum flood peak across China, to shed more light on the stationarity of annual maximum floods and the changing flood hazards over the country.

Our study is also motivated by Typhoon Nina and the resultant August 1975 flood in central China. The August 1975 flood in central China, with 26000 direct fatalities, is one of the most destructive floods in the world history (Yang et al., 2017). The August 1975 flood plays a key role in shaping the envelop curve of floods in China and different versions of the world envelop curve (Yang et al., 2017; Costa, 1987). The unit peak discharge is 17 $m^3 s^{-1} km^{-2}$ (which is peak discharge divided by drainage area) for a 760 $km^2$ drainage basin, and is on the list of the world maximum floods. The maximum 6-hour rainfall

accumulation of 830 mm is comparable to the world record (840 mm, Teegavarapu, 2013). Central China lies in the region with relatively less frequent visits of tropical cyclones in the western North Pacific basin (Wu et al., 2005; Jiang and Jiang, 2014). Previous studies show that landfalling tropical cyclones can make great contributions to extreme rainfall in inland regions, even though the frequency of occurrence is not as comparative as coastal regions (e.g., Zhang et al., 2018a). This is closely linked to a couple of factors, such as the interplay of tropical cyclone and baroclinic disturbances (i.e., known as extratropical transition,

Hart and Evans, 2000), interactions with mid-latitude systems (e.g., easterly, Shu et al., 2018; Feng and Shu, 2018), and impact of regional topography (as demonstrated by Typhoon Nina, Yang et al., 2017). The devastating consequences of Typhoon Nina and the August 1975 flood expose inadequacies of conventional approaches for flood frequency analysis (e.g., fitting historical flood records with assumed distribution function), and highlight the importance of hydrometeorological approaches for Probable Maximum Precipitation (PMP) / Probable Maximum Flood (PMF) analyses for better designs of flood-control

infrastructures (e.g., Smith and Baeck, 2015; Yang et al., 2017). In this study, we examine the impact of tropical cyclone on the upper tail properties of flood peak distribution over China. We provide characterizations of landfalling tropical cyclones that produced severe historical floods over China, focusing on the nature of the storm and relative locations of flood peaks to the circulation center. Results presented in this study can promote a predictive understanding of flood hazards associated with landfalling tropical cyclones. This is especially a critical issue over China due to its high frequency of landfalling tropical

cyclones (i.e., with 9.3 tropical cyclones making landfall on average per year) (Jiang and Jiang, 2014). Economic damages caused by landfalling tropical cyclones are rapidly increasing in recent decades across China, with a large portion of the damages resulted from extreme tropical cyclone rainfall and flooding (Zhang et al., 2009).

The rest of the paper is structured as follows. In section 2, we introduce the dataset of annual maximum flood peaks, followed by section 3 for detailed descriptions of methods, including change point and trend analyses, Generalized Extreme





Value distribution, and association of annual maximum floods with particular tropical cyclones. Results and discussions are provided in section 3, followed by summary and conclusions in section 4.

## 2    Data

Our analysis is based on observations of annual maximum instantaneous peak discharge from 1120 stream gauging stations with continuous records of at least 50 years. The longest flood record is 145 years. The dataset is comprehensively collected

from local hydrographic offices of nine major river basins across China. All these stations are nation-level control stations with little evidence of site re-location during the observational periods. Strict quality control procedures are implemented to ensure consistency and accuracy of the records. There are relatively more stations distributed in the eastern China than the western part of the country (Figure 1). The record length of 66% stations exceeds 60 years starting from 1950s till the year of 2017 (Figure 2a). There are considerable variabilities in the spatial scales of represented river basins, with a large percentage (approximately

64%) of stations representing small and medium river basins (with drainage areas less than 5000 km$^2$, Figure 2b). To facilitate analyses and comparisons, we further classify the 1120 stations into two sub-groups, i.e., northern and southern China, based on their geographic locations (Figure 1). The northern group includes stations mainly in northeastern river basins, the Yellow River basin, the Huaihe River basin, and the Haihe River basin, while the southern group includes southeastern river basins, southwestern river basins, the Yangtze River basin, and the Pearl River basin. Previous studies found contrasting climate

regimes and extreme rainfall distributions between northern and southern China (e.g., Yang et al., 2013; Ma et al., 2015).

## 3    Methodology

### 3.1    Change point and trend analysis

We use the nonparametric Pettitt's test (Pettitt, 1979) to examine the presence of abrupt changes in annual flood peak series. Pettitt's test is a rank-based test that relies on the Mann-Whitney statistic to test whether two samples come from the same

population. There are no assumed distributions for the test, which makes it less sensitive to outliers and skewed distributions. It allows for the detection of a single change point in mean at an unknown point in time, with the test significance computed using the given formulation. We further apply the Pettitt's test on the squared residuals derived with respect to the local polynomial regression line (loess function, Cleveland, 1979) to detect change point in variance in annual flood peak series (similarly see, e.g., Villarini et al., 2009; Villarini and Smith, 2010; Yang et al., 2013).

We use the nonparametric Mann-Kendall test (Mann, 1945; Kendall, 1975) to examine the presence of monotonically increasing or decreasing trends in annual flood peak series. For the series with change point in mean, we divide it into two sub-groups and test monotonic trends for each of the two sub-groups (i.e., before and after the change point). We assume the existence of only a single change point in mean for each flood peak series in this study, to avoid dividing the series into too many segments. We set a significance level of 5% for both the change-point and trend tests. Pettitt's test and Mann-Kendall



test are further applied in the series of timing of annual maximum flood peaks (represented by day of the year) to investigate changes in flood peak timing across China.

## 3.2  Generalized Extreme Value distribution

The Generalized Extreme Value (GEV) distribution is used to statistically model distributions of annual maximum flood peaks (e.g., Coles, 2001; Villarini and Smith, 2010). The GEV, based on extreme value theory, has been widely used in flood frequency

analysis (e.g., Coles, 2001; Katz et al., 2002; Morrison and Smith, 2002; Villarini and Smith, 2010). The cumulative distribution function of the GEV takes the form:

$$F(x|\mu,\sigma,\xi) = exp\left\{-[1+\xi(\frac{x-\mu}{\sigma})]^{-1/\xi}\right\} \tag{1}$$

where $\mu$, $\sigma$, and $\xi$ represents the location, scale, and shape parameter, respectively. The location ($\mu$) and scale ($\sigma$) parameter is related to the magnitude and variability of the records, respectively. The shape parameter ($\xi$) indicates the tail properties of the distribution, with positive (negative) values pointing to heavy and unbounded (light and bounded) upper tails of flood peak

distribution. The GEV parameters are estimated based on the maximum likelihood estimators (e.g., Coles, 2001). We fit the GEV distributions only for stations without statistically significant change points in mean and variance and monotonic trends.

We examine the dependence of GEV parameters on drainage area. Villarini and Smith (2010) examined annual flood peaks in the eastern US, and found a strong dependence of location and scale parameter on drainage area, while the shape parameter only shows a weak dependence on drainage area (Villarini and Smith, 2010). We further test whether or not drainage area can

explain the spatial variability of GEV parameters over China.

## 3.3  Association of flood peaks with tropical cyclones

We associate an annual flood peak of a given stream gauging station with a particular tropical cyclone following procedures in Villarini and Smith (2010) and Smith et al. (2011), i.e., if the center of a tropical cyclone is within 500 km of the gauging station during a time window of two weeks centered on the occurrence time of the flood peak. The thresholds (500 km and two

weeks) reflect the mean spatial extent of tropical cyclone rainfall and the upper limit of flood response time in the representing river basins. We obtain the information of tropical cyclones for the west Pacific basin from the International Best Track Archive for Climate Stewardship (IBTrACS, see https://www.ncdc.noaa.gov/ibtracs/ for details). The dataset provides records of the circulation center location (latitude and longitude) and storm intensity (represented by minimum sea level pressure) at a temporal interval of 6 hours. An additional attribute provided by IBTrACS for each tropical cyclone at each time interval is

the nature of the storm, i.e., extratropical transition (ET) or tropical storm (TS). Extratropical transition (ET) characterizes the changing properties of a tropical cyclone from a warm-core, symmetric structure to a cold-core, asymmetrical structure (e.g. Hart and Evans, 2000). Physical process associated with extratropical transition plays an important role in determining the spatial distribution of tropical cyclone rainfall (e.g. Hart and Evans, 2000; Atallah and Bosart, 2003; Atallah et al., 2007; Liu and Smith, 2016). Tropical storm (TS), as a contrast, indicates the maintenance of a warm-core, symmetric structure during

the entire life cycle of the storm.





## 4 Results and discussion

### 4.1 Stationarity for flood peak magnitude

Figure 3 shows the results of change-point analyses for annual flood peak magnitudes based on the Pettitt's test. There are 436 (38%) and 398 (35%) stations with significant change points in mean and in variance, respectively. 27% stations show

change points both in mean and in variance. The majority of stations tend to show smaller values in mean and/or variance after the change point (figure not shown). Change points in both mean and variance show striking spatial concentration in northern China (i.e., the lower Yellow River basin, the upper Huaihe River basin, and the entire Haihe River basin). Change points in both mean and variance are frequently observed during the period 1980-2000, with slightly larger frequency of occurrence during the period 1990-2000. We observe an additional amount of change points in mean distributed in the downstream of

southwestern river basins and in the upper and middle portion of the Yangtze River and Pearl River basins (Figure 3a). These change points tend to occur in the period 2000-2010 instead of the period of dominant change-point occurrence in northern China.

Spatial and temporal clustering of change points demonstrate evidence of anthropogenic influences on flood hydrology (e.g., Vogel et al., 2011; Hodgkins et al., 2019). We are able to relate some of the changes in annual flood peaks series to intentional

human activities. For instance, the change point in mean at the year of 1986 in the upper Yellow River, the Guide hydrological station, is due to the construction of a large hydropower-generation dam, the Longyangxia Dam (Figure 4a). The Longyangxia Dam is a multi-purpose dam (e.g., flood control, water supply), and controls runoff variability of the entire Yellow River basin (Si et al., 2019). The Guide station is approximately 30 km downstream of the Longyangxia Dam. There are a couple of other hydrological stations distributed further downstream (e.g., Xunhua hydrological station, 120 km downstream), and show

change points in mean around the year of 1986 for the annual flood peak series. Anthropogenic regulations on rivers in northern China (especially the middle/lower portion of the Yellow River basin and the upper Haihe River basin) is often characterized with a cascade construction of small reservoirs. We show a flood peak series in the upper Haihe River basin that experienced significant decrease in annual maximum flood peak magnitudes (smaller values both in mean and variance after the change point) around early 1990s, associated with extensive construction of small reservoirs due to increased demand for irrigation

and domestic water supply (Figure 4b). The impact of regulation by dams or reservoirs on flood hydrology has been discussed and debated in previous studies (e.g., Yang et al., 2008; Barros et al., 2014; Ayalew et al., 2017; Lu et al., 2018). For instance, Smith et al. (2010) found limited impacts of dams on flood hydrology in the Delaware River basin, which is not the case for the upper Yellow River basin in our study. This might be related to contrasting physiographic properties of the river basins and/or functions of the dams, and needs further analysis.

Changes in land use/land cover (e.g., urbanization, deforestation/afforestation) can also contribute to change points in the series of annual flood peaks. This is especially the case for stations in the lower Haihe River basin (where the Beijing-Tianjin-Hebei metropolitan region is distributed) and Yangtze River delta region (where Shanghai and other major cities are located). Figure 4c shows a small urban watershed in the lower Yangtze River basin) that experienced rapid urbanization in recent decades. Trans-boundary water transfer project demonstrates another form of anthropogenic influence on flood hydrology.





Abrupt increases in flood peak magnitudes are mainly tied to the elevated base flows transferred from neighbouring river basins. We provide the annual flood peak series for a station in the lower Yellow River basin (Figure 4d). Increasing water demand from domestic and agricultural sectors in the lower Yellow River basin lead to extensive implementation of water-transfer projects.

Abrupt changes in the series of annual flood peaks can also originate from the changes in extreme rainfall across China.
One of our previous studies investigated changes in annual maximum daily rainfall over China, but found no clear signature of spatial clustering for change points in either mean or variance for the rainfall series, although abrupt changes in annual maximum daily rainfall frequently occur in the 1990s (see Figure 2 in Yang et al., 2013). Inconsistent spatial patterns of change points in annual maximum flood peak and daily rainfall series may indicate a weak signal of climate shifts in producing abrupt changes in annual flood peaks.

We further examine the monotonic trends based on the Mann-Kendall test. There are only 69 stations (accounting for approximately 6% of the total stations) with significant linear trends (Figure 5a). For those stations with significant linear trends, 62 (7) of them exhibits decreasing (increasing) trends. The 62 stations are uniformly distributed across the entire country, indicating a weakening tendency of annual maximum flood peaks over China in recent decades. Abrupt change rather than slowly varying trend is a common mode of the violation of the stationarity assumption for the annual flood peak series
over China. For those stations with significant change points in mean, we test the liner trends for each sub-series of flood peaks before and after the change point. Almost all stations show decreasing trends for the sub-series either before or after the change point with only a few exceptions (Figure 5b and 5c). Similar with change points in mean and in variance, stations with significant decreasing trends after change points spatially concentrate in northern China, especially the middle and lower portion of the Yellow River basin and the upper Haihe River basin. The decreasing trend in the middle and lower portion of
the Yellow river is most likely due to the implementation of soil conservation practices in its tributary regions (e.g., Bai et al., 2016). There are few stations in southern China that show significant linear trends either before or after change points.

Changes in annual rainfall extremes (i.e., annual maximum daily rainfall) show a "dipole-like" spatial structure over China, with decreasing trends in northern China and increasing trends in the south (e.g., Yang et al., 2013; Ma et al., 2015; Gu et al., 2017b). The decreasing annual maximum flood peaks in northern China may be partially attributed to the weakening rainfall
intensity in recent decades. The opposite trends in annual rainfall extremes and annual maximum flood peaks in southern China seem contradictory to our perception. Contrasting trends between intense rainfall and annual high flows are also found over United States (mainly eastern of the Mississippi River), which are attributed to inconsistent changes of intense rainfall in different seasons (Small et al., 2006), i.e., changes in fall precipitation mainly contributes to the trend in annual rainfall extremes, while annual high flows are often observed in spring with no significant changes in rainfall. This is, however, not the
case for southern China. Changes in rainfall extremes among all four seasons are dominated by significant or relatively weak increasing trends over southern China (Gu et al., 2017b). Disconnections between changes in annual maximum rainfall and annual flood peaks are also identified in other previous studies (e.g., Ivancic and Shaw, 2015; Berghuijs et al., 2016; Wasko and Nathan, 2019), and point to the additional roles of antecedent watershed wetness and changes in space-time rainfall properties in dominating flood-generation processes (i.e., storm extent, Sharma et al., 2018). Disconnection of changes in rainfall extremes





and flood as exhibited for the gauges across southern China highlight the complex drivers for flood-generation process, and is worthwhile for further investigation.

Analysis on the stationarity of annual flood peaks across China point to mixed controls of human activities, external climate factors (i.e., extreme rainfall), and changes in soil moisture on flood hydrology. We note that further attribution analysis can provide additional insights into flood drivers and their changes, but can be challenging. The homogeneity of flood population
for flood frequency analysis need to be carefully revisited in a changing environment. This is especially proposed by England et al. (2018) in Hydrology Subcommittee Bulletin 17C as an imminent need to "define flood potentials for watersheds altered by urbanization, wildfires, deforestation, and by reservoirs". Our results highlight the importance of state-of-art process-based approaches (e.g., Wright et al., 2014; Yu et al., 2018) and statistical modeling approaches (Salas et al., 2018; Serago and Vogel, 2018; Gao et al., 2019; Dong et al., 2019; Barth et al., 2019) for flood frequency analyses across China, especially for northern
China that exhibits an overwhelming frequency of stations with nonstationarities.

### 4.2 Seasonality of annual flood peaks

We examine seasonal distribution of annual flood peaks to highlight the mixture of flood-generating systems over China. There are three (two) distinct peaks in the seasonal distribution of annual flood peaks for southern (northern) China (Figure 6). The first peaks for both southern and northern China occur around late April, but are resulted from different flood-generating
systems. Frequent occurrence of annual flood peaks around late April in southern China are observed mainly in the southeastern coast, and are caused by frontal systems or associated with early onset of the East Asia Summer Monsoon. The April peak of flood frequency in northern China is contributed by localized storm events associated with mid-latitude weather systems in the northwestern part of the country, or related to snow melt in high-altitude regions (Ding and Zhang, 2009). The East Asia Summer Monsoon onsets around early May over mainland China, and moves stepwise northward/northeastward driven by the
West Pacific Subtropical High (e.g., Ding and Chan, 2005; Zhang et al., 2017). The monsoon system is characterized with "two abrupt northward jumps and three stationary periods", and plays a deterministic role in the seasonal distribution of flood peaks in both northern and southern China. Frequent flood peaks around late June in the middle and lower portion of the Yangtze River basin contribute to the second peak of seasonal distribution of flood frequency in southern China. Further northward propagation of the monsoon system leads to frequent annual flood peaks in northern China around late July and early August.
The summer monsoon retreats back to the south and is weakened afterwards, transferring the dominance in flood-generating systems to tropical cyclones and post-monsoon synoptic systems.

Annual flood peaks that are caused by tropical cyclones show a very sharp seasonal distribution, with 70% of them observed in August alone (Figure 6). Strong pressure gradients along the western flank of the West Pacific Subtropical High provide favorable synoptic conditions for large-scale moisture transport and north-westward propagation of tropical cyclones. Inter-
actions of tropical cyclones with mid-latitude systems and regional topography (i.e., Qinling and Tainhang Mountains) can further enhance tropical cyclone rainfall and the resultant flooding over eastern China (e.g., Svensson and Berndtsson, 1996; Yang et al., 2017; Gu et al., 2017a). The seasonal distribution of annual flood peaks in northern China is almost overlapped with that of flood peaks caused by tropical cyclones, while tropical cyclones mainly contribute to the third peak of the sea-





sonal distribution for annual flood peaks in southern China (Figure 6). The concurrency of monsoon-controlled storm events
and tropical cyclones is a key element of flood hydrometeorology and hydroclimatology in eastern China. As we will further
demonstrate in section 4.4, even though landfalling tropical cyclones are relatively more frequent in southern China, annual
flood peaks caused by relatively infrequent tropical cyclone visits play a vital role in determining the upper tail properties
of flood peak distributions in northern China. Monsoon-related extreme rainfall dominates the upper tail of flood peaks in
southern China.

Figure 7 shows stations with significant change points in mean and monotonic trends for the series of annual flood peak
timing. Compared to flood peak magnitude, flood peak timing exhibits weak decadal variations. There is a considerably small
number of stations with significant change points in mean (and in variance, figure not shown) for flood peak timing. For those
stations with significance, they show similar spatial concentration with that for flood peak magnitude (Figure 7a). Abrupt
changes in flood peak timing tend to occur during the period 1980-1990, consistent with what we previously found for the
annual maximum daily rainfall series over China (Yang et al., 2013). There is a notable spatial split in terms of monotonic trends
for flood peak timing, with decreasing (increasing) trends in northern (southern) China (Figure 7b). Villarini (2016) found
limited impact of urbanization and river regulations on the average timing of flooding across the continental United States,
even though the strength of seasonality is weakened. Delayed occurrence of annual flood peaks is consistent with changes
in the seasonality of heavy precipitation across China related to the delayed occurrence of annual maximum daily rainfall in
southern China (Gu et al., 2017b). Previous studies show that later onset of East Asia Summer Monsoon and intensified rainfall
during the monsoon and post-monsoon season might have leads to the changing seasonality of extreme rainfall over China (Day
et al., 2018). Changes in flood peak timing can be resulted from changes in both rainfall and antecedent soil moisture, but are
not necessarily related to changes in annual peak rainfall. Contrasting changes in the timing of annual flood peaks point to
a tendency for a more centralized seasonal distribution of annual flood peaks over the entire country through minimizing the
current shift of seasonal distributions between northern and southern China (as presented in Figure 6). Changes in the flood peak
timing highlight the necessity of revisiting operation rules for multi-objective dams (i.e., flood-control, hydropower, irrigation,
water supply) across China. For instance, reservoir managers in southern China possibly need to consider to delay the release of
water storage in reservoirs so that sufficient water be maintained for irrigation and domestic water use. A centralized seasonal
distribution of annual flood peaks calls for coordinated flood-control practices across the entire country.

## 4.3 Generalized Extreme Value distribution

We model distributions of annual flood peak magnitudes using the GEV distribution. We only focus on the stations without
significant change points in mean or in variance, and without significant monotonic trends. There are 486 stations that satisfy
these requirements. These stations are densely located in southern rather than northern China (Figure 8), mostly due to the spa-
tial clustering of stations with abrupt change points in annual flood peaks in northern China (Figure 3). The stationary stations
represent a wide range of spatial scales of river basins for both northern and southern China. Figure 9 shows the dependence
of GEV parameters on drainage area for the 486 stationary stations. Location and scale parameters are positively correlated
with drainage area in a log-log domain. The correlations are all significant at the level of 5%. The shape parameter, however,





generally decreases with drainage area but shows only weak dependence in a log-log domain (no statistical significance). The upper tail properties (as represented by the shape parameter) of flood peak distributions are weakly determined by drainage

area, while the magnitude and variability of annual flood peaks can be well explained by drainage area. Our results are consistent with the study in the eastern US by Villarini and Smith (2010), and contribute to generalized understandings on the upper tails of flood peak distributions.

An interesting finding for annual flood peaks over China is that there are striking spatial splits in terms of the dependence of the GEV parameters on drainage area between northern and southern China (Figure 9). The location and scale parameters for

stations in southern China are consistently larger than their counterparts in the north (with a few exceptions, Figure 9a and 9b). The shape parameters in northern China are comparatively larger than that in southern China. Large shape parameters indicate heavier upper tails of flood peak distributions in northern than southern China, even though the magnitudes and variability of flood peaks are relatively smaller in the north. One of our previous studies on the distribution of annual maximum daily rainfall found similar spatial splits for the dependence of GEV parameters on elevation between northern and southern China (Yang

et al., 2013) (see also a most recent study by Gu et al., 2017a). Spatial splits in extreme rainfall distributions highlight spatial heterogeneity in flood hydroclimatology across China. Spatial contrasts of extreme rainfall distribution further lead to different relations between the three GEV parameters and drainage area for flood peak distributions between northern and southern China. Regional flood frequency analysis should explicitly address the spatial splits through considering spatial heterogeneity in flood hydroclimatology.

We further show the spatial splits for the shape parameter in Figure 8. The majority of the northern stations show positive shape parameters, while southern stations are mixed with both negative and positive shape parameters. Contrasting space-time rainfall organizations seems to be a more effective predictor in explaining the spatial variability of the shape parameter than drainage area. Our results highlight the importance of hydrometeorological analyses for better characterizations of the upper tail properties of flood peak distributions (similarly see e.g., Smith and Baeck, 2015; Yang et al., 2017). Positive shape

parameters in northern China indicate flood peak distributions with unbounded upper tails, while negative shape parameters for most southern stations show flood peak distributions with upper bounds. Understandings remain poor pertaining to the nature of the upper tails of flood peak distributions (e.g., Smith et al., 2018). The bounded upper tails of flood peak distributions in the south can be associated with physical constrains over river basins (for instance, large dams for flood-control purposes) and/or the upper bounds to the hydroclimatological processes (see, e.g., Enzel et al., 1993; O'Connor et al., 2002; Serinaldi

and Kilsby, 2014).

### 4.4    Tropical cyclones and upper tail properties

In this section, we focus on tropical cyclones and their impacts on the upper-tail properties of flood peak distributions over China. Tropical cyclones contribute to approximately 18% of annual flood peaks over China. Figure 10 shows the map of the percentage of annual flood peaks that are caused by tropical cyclones to total annual flood peaks for each station. More than

50% of the annual flood peaks are caused by tropical cyclones in the southeastern coast of China, with the percentage even attaining 90% over the Hainan Island. The percentage gradually decreases when we move further inland and to higher latitudes.





Less than 10% annual flood peaks can be associated with landfalling tropical cyclones in the middle portion of the Yellow River and Yangtze River basins (Figure 10). The percentage of annual flood peaks caused by tropical cyclones is closely tied to the spatial distribution of tropical cyclone rainfall and frequency of tropical cyclone occurrence over China (Wu et al., 2005; Ren et al., 2010; Gu et al., 2017b). More than 30% of the extreme rainfall events are induced by tropical cyclones along the coastal regions (Gu et al., 2017a, b), with the percentage gradually decreased moving inland due to rapid weakening of storm intensity (e.g., surface roughness, insufficient moisture transport).

We further show the stations with record floods (i.e., the largest flood peak for the entire record of a station) that are caused by tropical cyclones in Figure 10 to highlight the impacts of tropical cyclones on the most extreme floods. Stations with record floods caused by tropical cyclones are spatially clustered in the southeastern coast, central and northeastern China (Figure 10). Tropical cyclone-induced record floods in the southeastern coast are mainly associated with abundant moisture and energy supply for extreme rainfall right after tropical cyclones making landfall. However, the spatial clustering of record floods by tropical cyclones in northern China (more specifically, the upper Huaihe River and northeastern China) can be partially related to extratropical transition processes during the life cycle of the storm and/or interactions with regional topography (i.e., Taihang and Qinling Mountains), as will be elaborated below. We do not observe a comparable distribution of record floods caused by tropical cyclones in southern China (e.g., the Yangtze River basin) excluding the coastal regions, even though the percentage of annual flood peaks caused by tropical cyclone is comparable to that in northern China (less than 30%, Figure 10). Our results highlight the impacts of tropical cyclones on flood peak distributions in northern China with a large percentage of record floods caused by relatively infrequent visits of landfalling tropical cyclones.

The impacts of tropical cyclones on the upper tail properties of flood peak distributions are further examined through the shape parameter of the GEV distribution. We compare the shape parameters between the entire annual flood peak series and the series with annual flood peaks caused by tropical cyclones removed (Figure 11). We focus on the series with record length exceeding 30 years after annual flood peaks caused tropical cyclones being removed from the series. This leads to the exclusion of most stations in the southeastern coast due to the high percentage of tropical cyclone-induced annual flood peaks (Figure 10). As can be seen from Figure 11, the scatters are generally distributed along the 1:1 line, indicating overall small changes in the shape parameters between two series. However, if we restrict our attention to the stations with record floods caused by tropical cyclones (mainly those stations in northern China), we observe significantly smaller shape parameters (see the insert box plot in Figure 11) for the series with annual flood peaks caused by tropical cyclones removed. Smaller shape parameter implies a lighter tail of flood peak distribution. Small variations in the shape parameters as demonstrated for the rest of the stations indicate relatively weak impacts of tropical cyclones on the upper tail properties of flood peak distributions. These stations are mainly located in inland regions of southern China. Our results are different from the study of Villarini and Smith (2010) in eastern United States that shows significant decreases in shape parameters for the majority of stations when annual flood peaks caused by tropical cyclones are removed from the series. The differences are tied to contrasting flood-generation mechanisms between China and the eastern US. Tropical cyclones and extratropical systems play central roles in the mixture of flood-generation mechanisms for the flooding in the eastern US (Smith et al., 2011). Extreme rainfall associated with East Asia Summer Monsoon, rather than landfalling tropical cyclones, can be a more important player in characterizing the upper tails





of flood peak distributions in most inland regions of southern China (e.g., the middle and lower portion of the Yangtze River basin) (Zhang et al., 2017). Tropical cyclones in northern China, even though characterized with low frequency of occurrence, pose significant influences on the upper tail properties of flood peak distributions. Contrasting roles of tropical cyclones in flood

peak distributions highlight the necessity of tailored procedures for flood-control practices and flood hazard assessment across China. For instance, landfalling tropical cyclones can be good candidates for PMP/PMF designs for river basins in northern rather than southern China.

We focus on tropical cyclones that produced relatively large numbers of flood peaks over China, to shed light on the physical attributes of flood hazards associated with landfallling tropical cyclones. There are 9 tropical cyclones that produced more than

100 annual flood peaks over China since late 1950s till present. The 9 tropical cyclones alone contribute to approximately 50% of total annual flood peaks caused by tropical cyclones. Figure 12 and Table 1 provide a summary of the 9 tropical cyclones. Typhoon Herb (1996) produced the largest number of annual flood peaks (167 in total), followed by Typhoon Wendy (1963) and Typhoon Tim (1994). Typhoon Herb (1996) produced a large number of annual flood peaks right after its landfall in mainland China (Figure 12a). Almost all the annual flood peaks caused by other tropical cyclones are distributed over the most

inland regions (Figure 12).

The 9 tropical cyclones can be further categorized into two groups according to the nature of the storm and spatial patterns of their tracks. The first group includes Typhoon Herb (1996), Typhoon Andy (1982), and Typhoon Nina (1975). The three tropical cyclones did not experience extratropical transition during the entire life cycle of the storms, and are characterized with two landfalls (i.e., Taiwan and mainland China). The tracks of these three tropical cyclones do not fall into the prevailing

tropical cyclone tracks in the Western North Pacific basin (Wu et al., 2005). Typhoon Nina (1995) produced the largest number of record floods (24 in total) among all historical tropical cyclones over China, followed by Typhoon Polly (1960) (14 in total) and Typhoon Andy (1982) (10 in total). Annual flood peaks and record floods caused by tropical cyclones in the first group are frequently observed in northern China (mainly the middle portion of the Yellow River and the upper Huaihe River basins). This region is characterized with complex terrain (i.e., Taihang and Qinling Mountains). Interactions of tropical cyclones with

regional topography can significantly enhance rainfall intensity through orographic lifting, as demonstrated by Typhoon Nina (1975). For instance, historical records of extreme rainfall (e.g., three-day rainfall accumulation exceeding 1000 mm) from Typhoon Nina (1975) were observed in the windward topographic regions (Yang et al., 2017). The other 6 tropical cyclones in Table 1 are categorized into the second group (Figure 12). A common feature for the tropical cyclones in the second group is extratropical transition process during the life cycle of the storms. Annual flood peaks are frequently observed after the

extratropical transition process (see the curvatures of tropical cyclone tracks in the latitudes around 30° in Figure 12), and are frequently observed in northern China. Except Typhoon Herb (1996), 4 of the top 5 largest number of annual flood peaks are caused by tropical cyclones with extratropical transition.

There are no strong preferences for the spatial distribution of annual flood peaks with respect to storm tracks (i.e., left or right of the track), even though the records floods caused by tropical cyclones tend to be frequently observed in the left-

front quadrant (typically the down-shear side) of the circulations. This is related to the preferable distribution of extreme tropical cyclone rainfall, due to enhanced moisture convergence and updraft on the down-shear side of the circulation (e.g.,





Atallah et al., 2007; Shu et al., 2018). Future studies need to investigate variabilities in the physical properties of river basins (i.e., drainage area, slope, shape) and their relationships with flood peaks (i.e., frequency and magnitude) caused by tropical cyclones, to shed more light on flood hazards associated with landfalling tropical cyclones over China.

## 5 Conclusions

In this study, we examine flood peak distributions over China based on 1120 stream gauging stations with continuous records of annual maximum instantaneous discharge for more than 50 years. The principal findings of this study can be summarized as follows.

(1) There are 38% and 35% stations exhibiting significant change points in mean and in variance, respectively. Change
points tend to occur during the period 1980-2000, and show strong spatial concentration in the lower Yellow River, upper Huaihe River, the entire Haihe River, upper Yangtze and Pearl River basins. Hydrological regimes in these regions demonstrate intensive anthropogenic influences, for instance, large hydropower generation dam, cascade constructions of small-capacity reservoirs, trans-boundary water transfer projects, soil-water conservation projects, urbanization. There is a weak signal of climate impacts on the abrupt changes in annual flood peaks across China. Abrupt change is the dominant mode of the violation
of stationary assumption for annual flood peaks over China.

(2) Approximately 6% stations (69 in total) show significant linear trends in the annual flood peak series. Those stations with significant trends are uniformly distributed across the country, with 62 of them exhibiting significantly decreasing trends. The decreasing trends of flood peak magnitude in northern China may be at least partially tied to changes in extreme rainfall. Disconnections between changes in annual rainfall extremes and annual maximum floods are identified in southern China, and
highlight complex flood-generation processes across China. The dominance of decreasing trends in annual flood peak series indicates weakening tendencies of severe flood hazards (i.e., annual maximum floods) over China, even though flood-affected area and economic damages are on the rise in recent decades (Kundzewicz et al., 2019). Future studies need to further examine changes in flood frequency for a complete assessment on flood hazards (based on peaks-over-threshold flood series, similarly see, e.g., Mallakpour and Villarini, 2015).

(3) Flood-generation systems over China show a mixture of East Asia Summer Monsoon, tropical cyclones, and extratropical systems. There is a temporal shift in the seasonal distribution of flood peaks between northern and southern China. Compared to flood peak magnitude, there are fewer stations exhibiting significant change points and/or linear trends in flood peak timing. For those stations with significant linear trends in flood peak timing, the decreasing trends tend to occur in northern China, while the opposite is true for southern China. Changes in flood peak timing tend to minimize the shift of seasonal distribution
of annual flood peaks between northern and southern China, leading to centralized seasonality of annual maximum floods over China.

(4) We fitted GEV distribution for the stationary time series of annual flood peaks, and examined the dependence of its parameters on drainage area. We found that the location and scale parameters are linearly scaled with drainage area in a log-log domain. There is only a weak tendency for the shape parameters to decrease as a function of drainage area. Our results are con-





sistent with previous studies, and highlight scale-independent properties of upper tails of annual flood peaks. The relationships between GEV parameters and drainage area show strong spatial splits between northern and southern China, indicating space-time rainfall organization as an important player in determining the upper-tail properties of flood peak distributions. Procedures for regional flood frequency analysis over China should explicitly address spatial heterogeneity in flood hydroclimatology.

(5) Tropical cyclone plays an important role in characterizing spatial-temporal variability of flood peaks and the upper-tail
properties of flood peak distributions over China. More than 50% of the annual flood peaks in the southeastern coast are caused by tropical cyclones. The percentage progressively decreases when we move further inland and to higher latitudes. Tropical cyclones lead to heavier tails of flood peak distributions (with larger shape parameters of the GEV distribution) in northern China. Those regions are characterized with record floods frequently associated with tropical cyclones, despite that tropical cyclone visits relatively infrequently compared to the southern China. Record floods in southern China are more
frequently associated with monsoon-related extreme rainfall events rather than tropical cyclones. We highlight the importance of considering the mixture of flood-generating mechanisms in flood frequency analyses especially in northern China.

(6) There are 9 tropical cyclones that produced more than 100 annual flood peaks over China. The 9 tropical cyclones contribute to approximately 50% of total annual flood peaks caused by all historical tropical cyclones. Large number of annual flood peaks is associated with extended spatial coverages of extreme rainfall after the storms going through extratropical
transition. It can also be tied to favorable synoptic set-up for persistent moisture transport after the storm making landfall, as demonstrated by Typhoon Herb (1996), Typhoon Andy (1982), and Typhoon Nina (1975). Interaction of tropical cyclone with regional topography (i.e., Taihang and Qinling Mountains) is a key element for severe flood hazards in central China (mainly the middle/lower Yellow River basin and upper Huaihe River basin). Annual flood peaks caused by tropical cyclones do not show strong spatial preference with respect to the track, even though the record floods tend to be frequently observed in the left-
front quadrant of the circulation. Typhoon Nina (1975) produced the largest number of record floods, and plays a critical role in shaping the envelope curve of floods over China. Hydrometeorological analyses can provide improved characterization on the physical attributes of flood hazards associated with landfalling tropical cyclones (e.g., Yang et al., 2017). Previous studies show strong teleconnections between the activities of tropical cyclones over the western North Pacific basin and large-scale atmospheric forcing, e.g., the El Niño-Southern Oscillation (ENSO, e.g., Chan and Shi, 1996; Chan, 2000), Madden-Julian
Oscillation (MJO, Kim et al., 2008). Future studies need to investigate the linkage between tropical cyclone floods and remote atmospheric forcing (similarly see, e.g., Aryal et al., 2018) to better understand decadal changes in flood hazards associated with landfalling tropical cyclones over China.

*Data availability.* The data used in this research are collected from distributed hydrological offices of major river basins over China. The dataset is unavailable to access due to licensing issues at the moment.





*Author contributions.* LY designed the study and carried out the analysis. LY wrote the manuscript with the contribution of LW. All authors contributed to the discussion and revision.

*Competing interests.* The authors declare that they have no conflict of interest.

*Acknowledgements.* This research is supported by the Strategic Priority Research Program of the Chinese Academy of Sciences (XDA230402). LX acknowledges support from the National Science Foundation of China (51609256) and the Young Elite Scientists Sponsorship Program
by the China Association for Science and Technology (2017QNRC001). The authors would like to extend sincere thanks to colleagues and collaborators from hydrographic offices of major river basins for their contribution to the dataset.



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





**Table 1.** Summary of tropical cyclones that produced more than 100 annual flood peaks over China. The "storm type" column shows whether the tropical cyclone experienced extratropitcal transition (ET) or not (TS).

| Rank | Name | No. of annual flood peaks | No. of record floods | Storm type |
|---|---|---|---|---|
| 1 | Herb (1996) | 167 | 4 | TS |
| 2 | Wendy (1963) | 159 | 6 | ET |
| 3 | Tim (1994) | 156 | 2 | ET |
| 4 | Freda (1984) | 144 | 2 | ET |
| 5 | Doris (1961) | 119 | 2 | ET |
| 6 | Winnie (1997) | 114 | 0 | ET |
| 7 | Andy (1982) | 111 | 10 | TS |
| 8 | Russ (1994) | 104 | 1 | ET |
| 9 | Nina (1975) | 102 | 24 | TS |

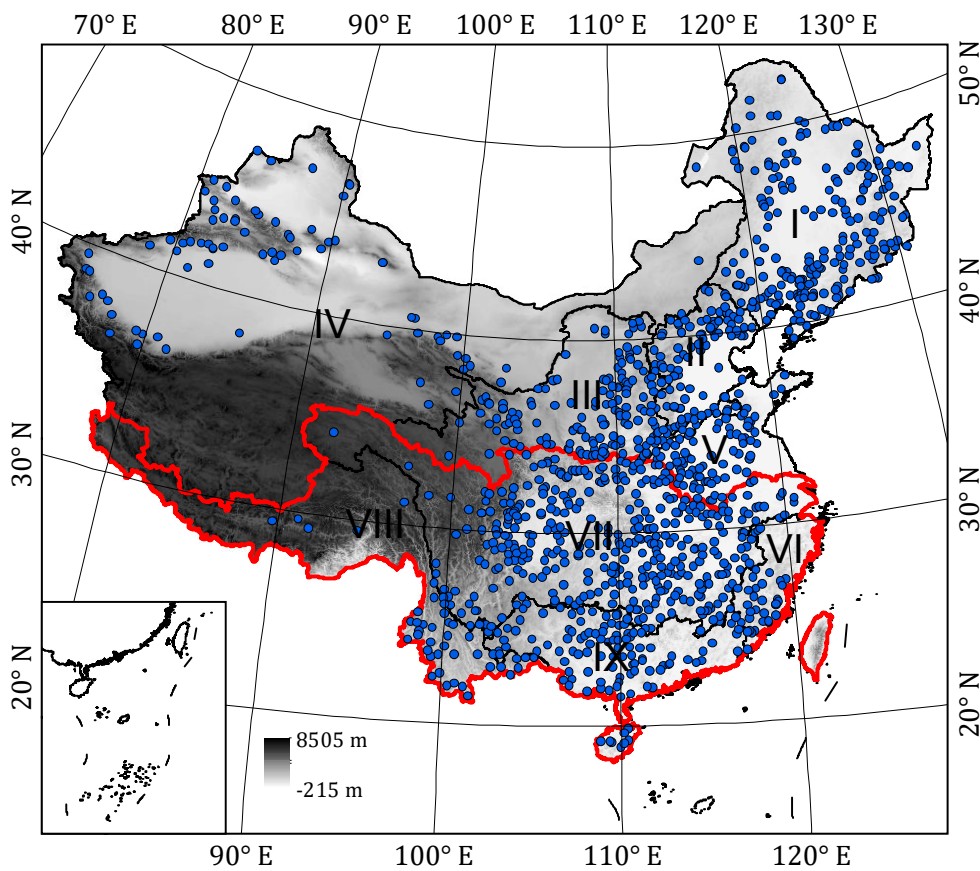

**Figure 1.** Overview of the stream gauging stations (blue dots) with record lengths of more than 50 years over China. Shaded color represents topography, while the black lines represent the first-level hydrologic units. The Roman numerals highlight the nine major hydrologic units in China: I-Northeastern river basins, II-Haihe River basin, III-Yellow River Basin, IV-Northwestern river basins, V-Huaihe River basin, VI-Southeastern river basins, VII-Yangtze River basin, VIII-Southwestern river basin, and IX-Pearl River basin. Red line shows the boundary of river basins in southern China (VI-IX), with the rest of the river basins in northern China (I-V).





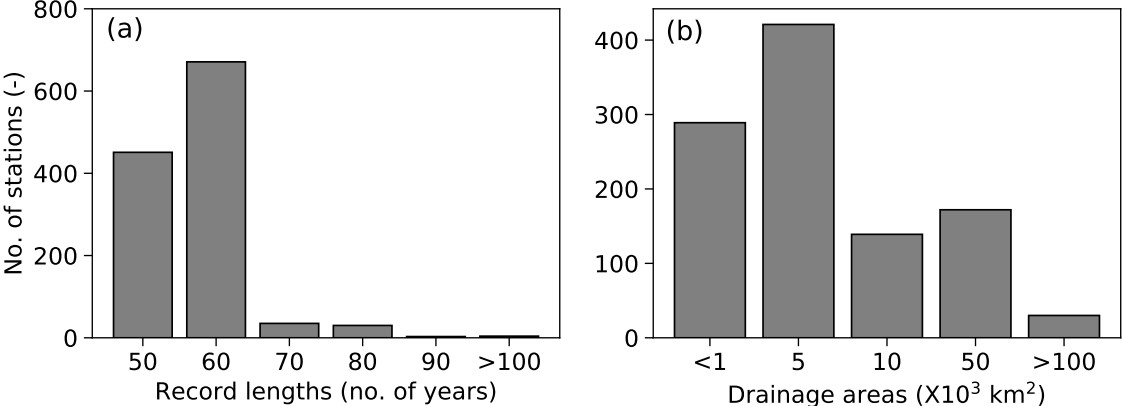

**Figure 2.** Histograms of stream gauging stations sorted by (a) record lengths and (b) drainage area.

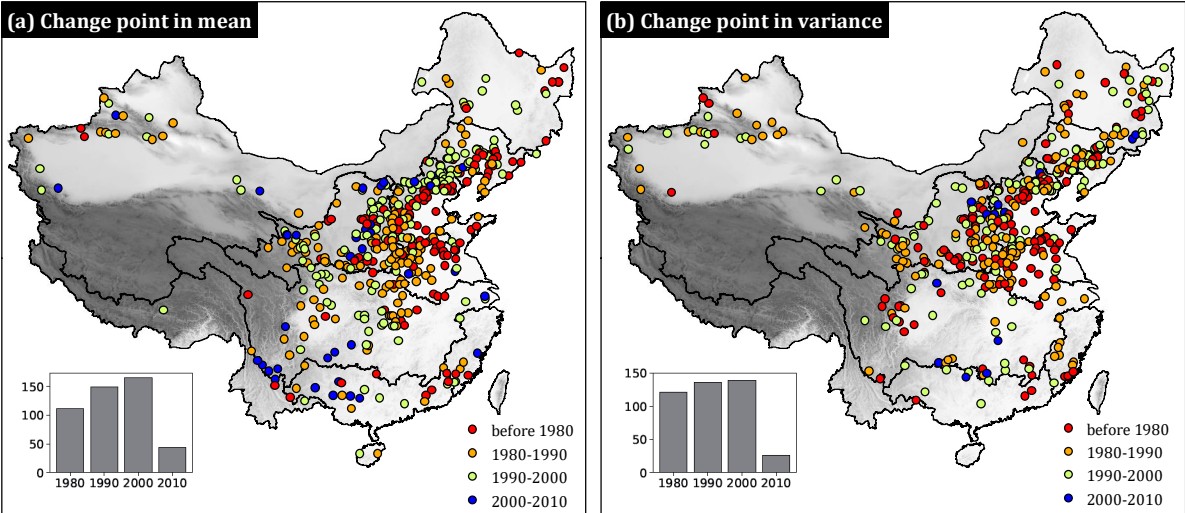

**Figure 3.** Change points in (a) mean and (b) variance. Color represents the year of change-point occurrence. The insert plot shows the histogram of the years of change-point occurrence (y-axis represents the number of change points, while x-axis represents year). Results are statistically significant at the level of 5%.



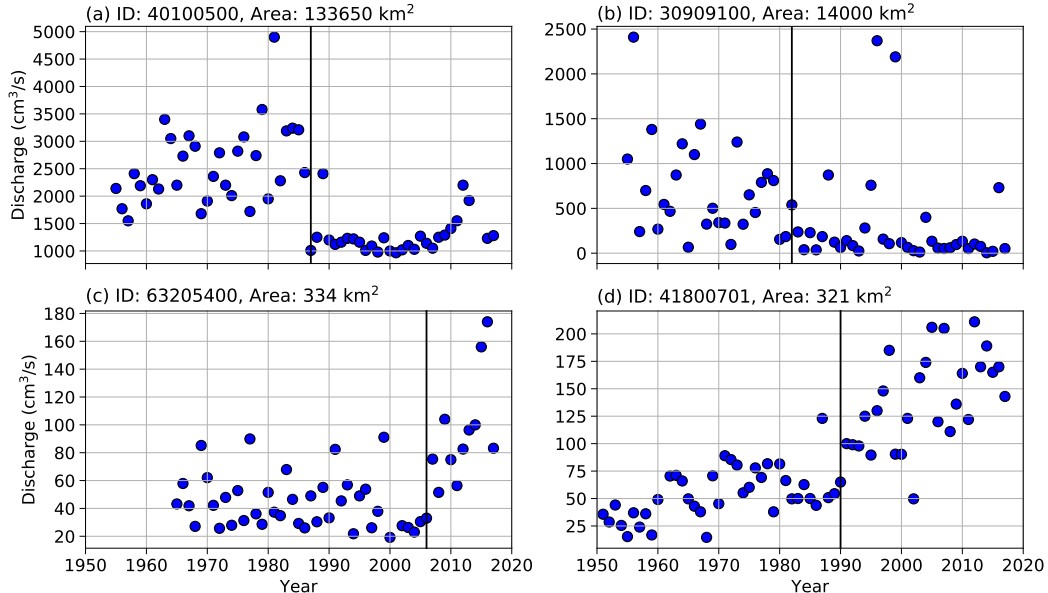

**Figure 4.** Time series of annual flood peaks for four stream gauging stations with strong human interventions: (a) large hydroelectric dams (upper Yellow River, ID: 40100500, 36.00$^o$N, 101.40$^o$E), (b) a cascade of small reservoirs (upper Haihe River, ID: 30909100, 38.39$^o$N, 113.71$^o$E), (c) urbanization (a tributary in the lower Yangtze River, ID: 63205400, 31.20$^o$E, 120.66$^o$E), and (d) water transfer project (a tributary in the lower Yellow River, ID: 41800701, 36.71$^o$N, 117.07$^o$E). Black lines indicate the year of occurrence for change point in mean. Results are based on the Pettitt's test, and are statistically significant at the level of 5%.

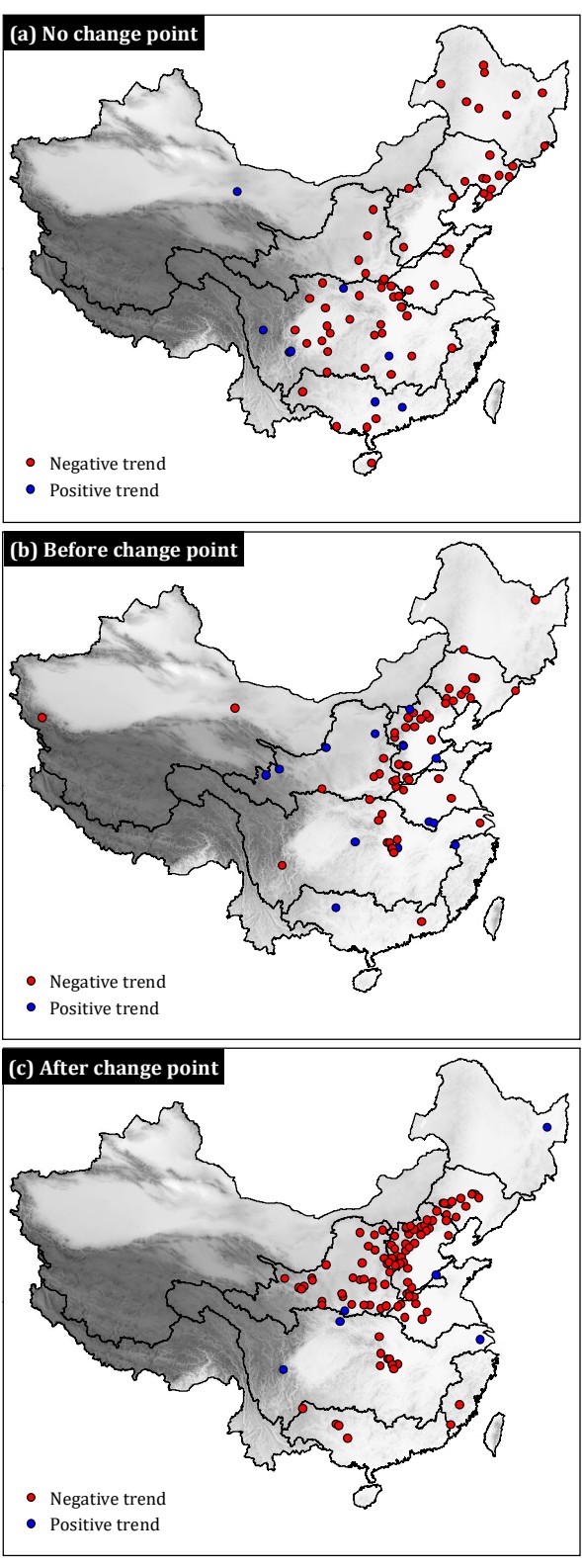

**Figure 5.** Mann-Kendall test results for stations (a) without change point in mean and (b,c) with change point in mean. Results are statistically significant at the level of 5%.





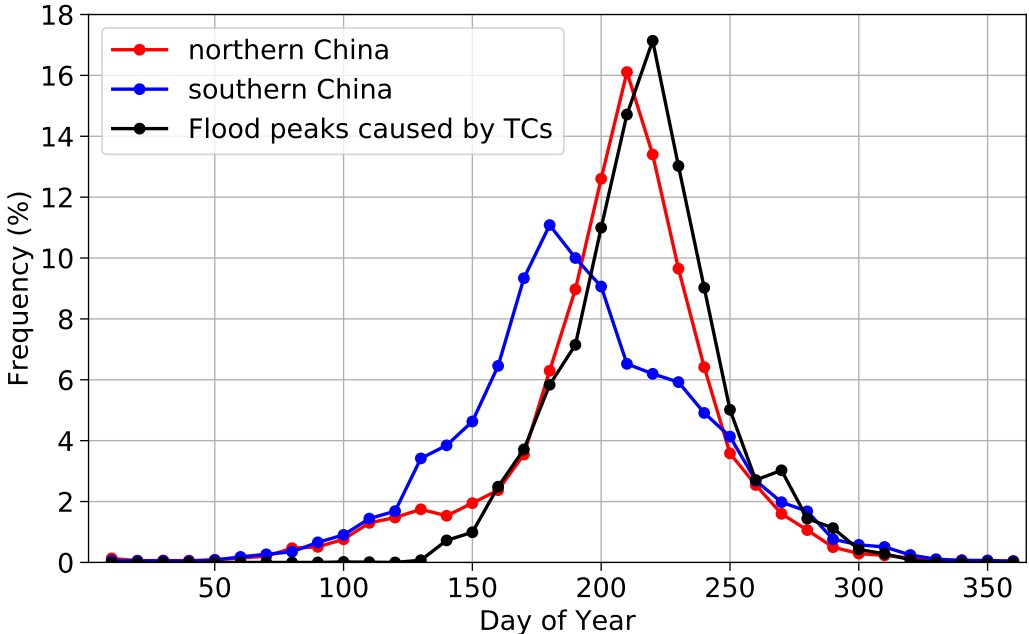

**Figure 6.** Seasonality of annual maximum flood peaks for northern China (red), southern China (blue), and annual flood peaks caused by tropical cyclones (black).





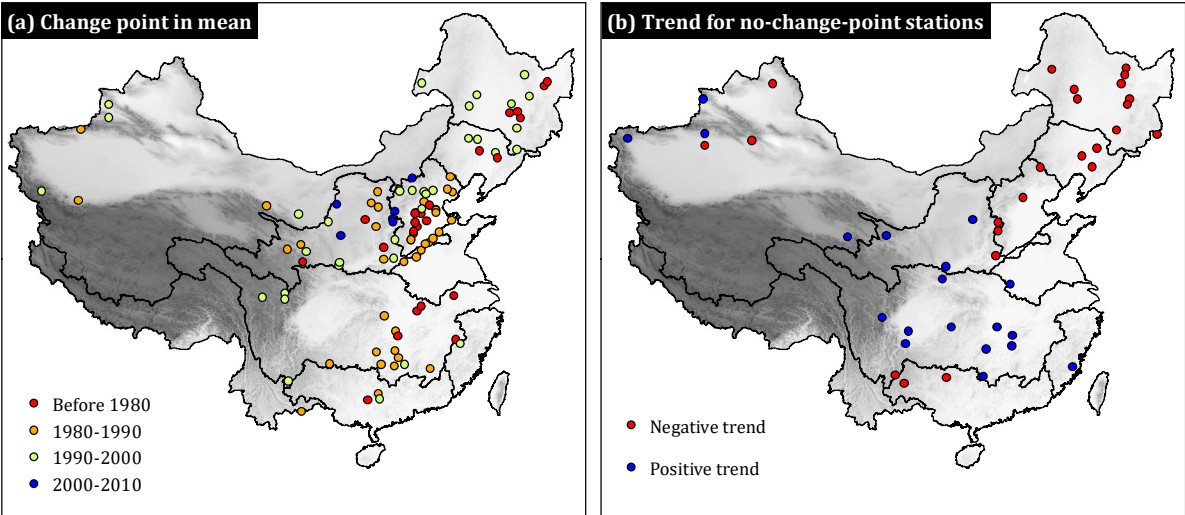

**Figure 7.** (a) Change point in mean for time series of annual flood peak timing (represented by day of the year). Color represents the year of change-point occurrence. (b) Mann-Kendall test for stations without change point in mean for the annual flood peak timing. Results are statistically significant at the level of 5%.



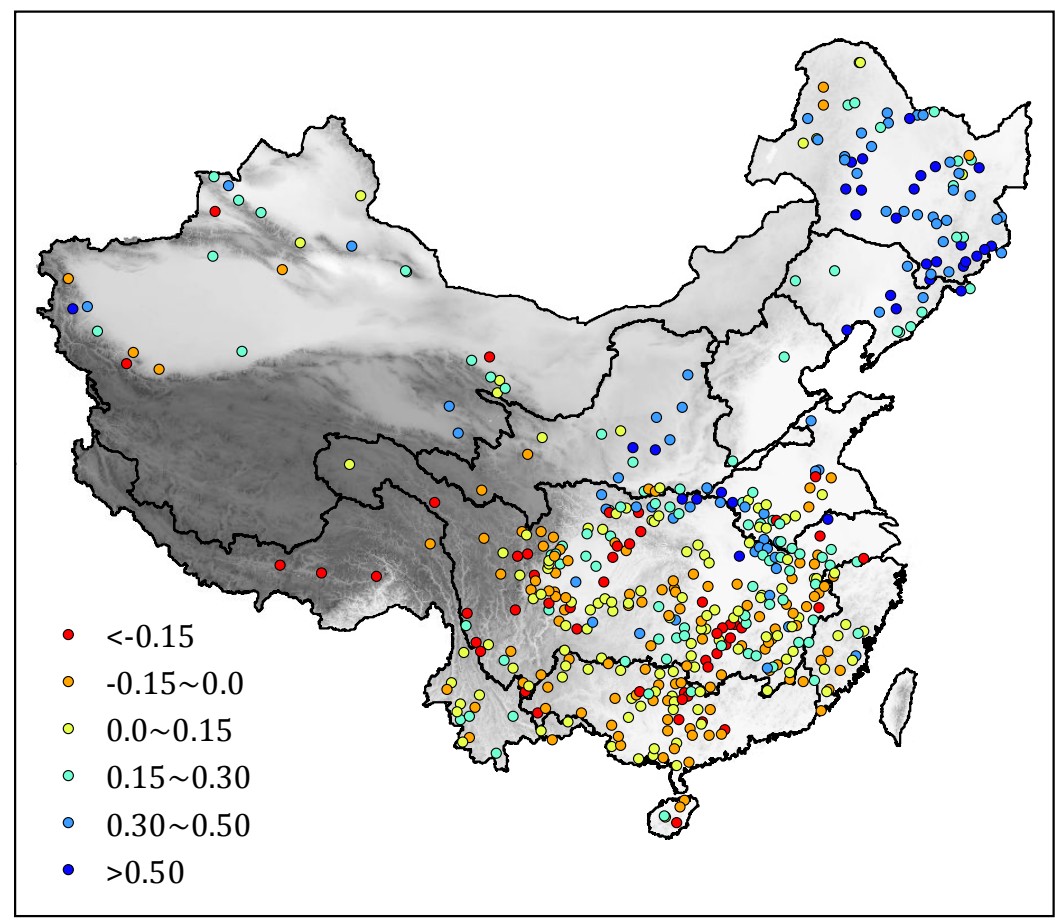

**Figure 8.** Map of the GEV shape parameters for the stationary time series of annual flood peaks.



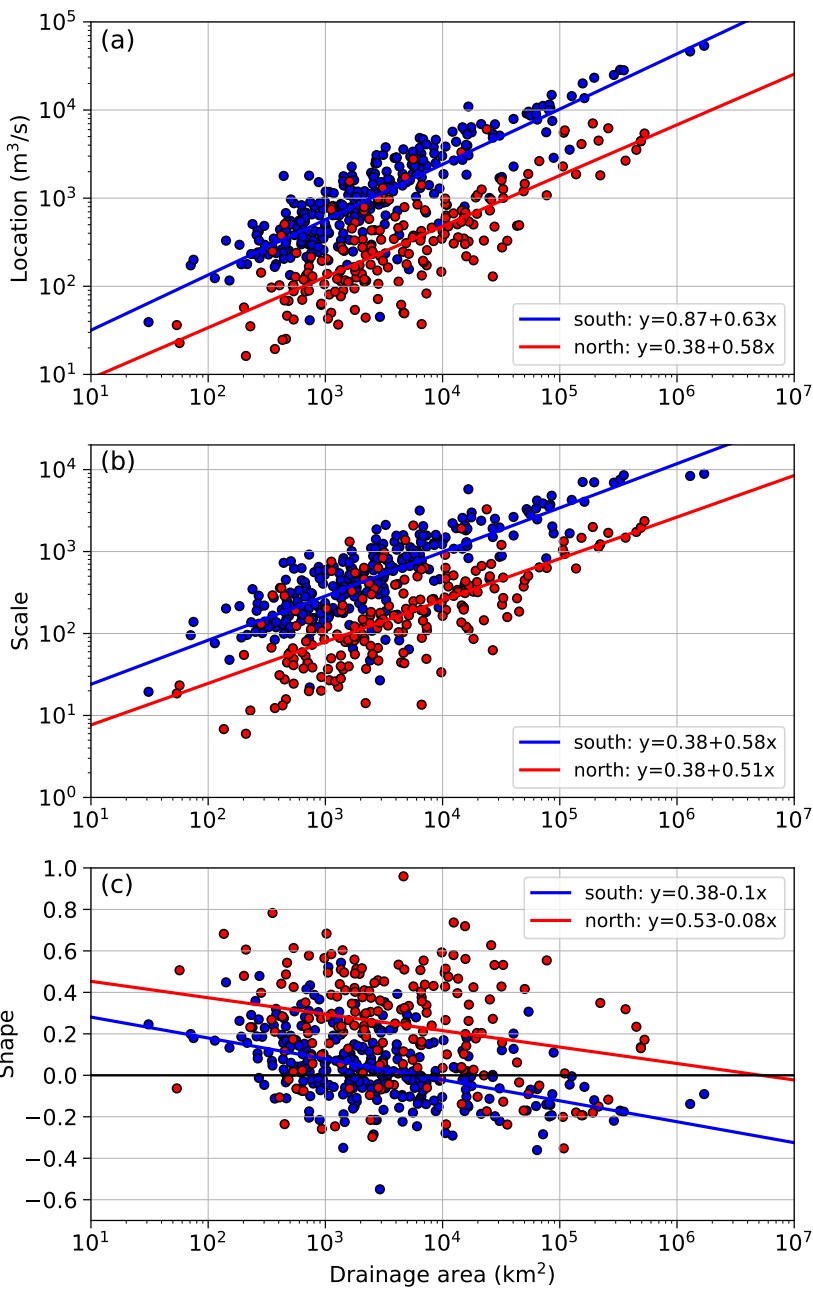

**Figure 9.** Scatterplots of GEV parameters (a) location, (b) scale, and (c) shape, as a function of drainage areas. Blue (red) scatters represent stations over south (north) China.

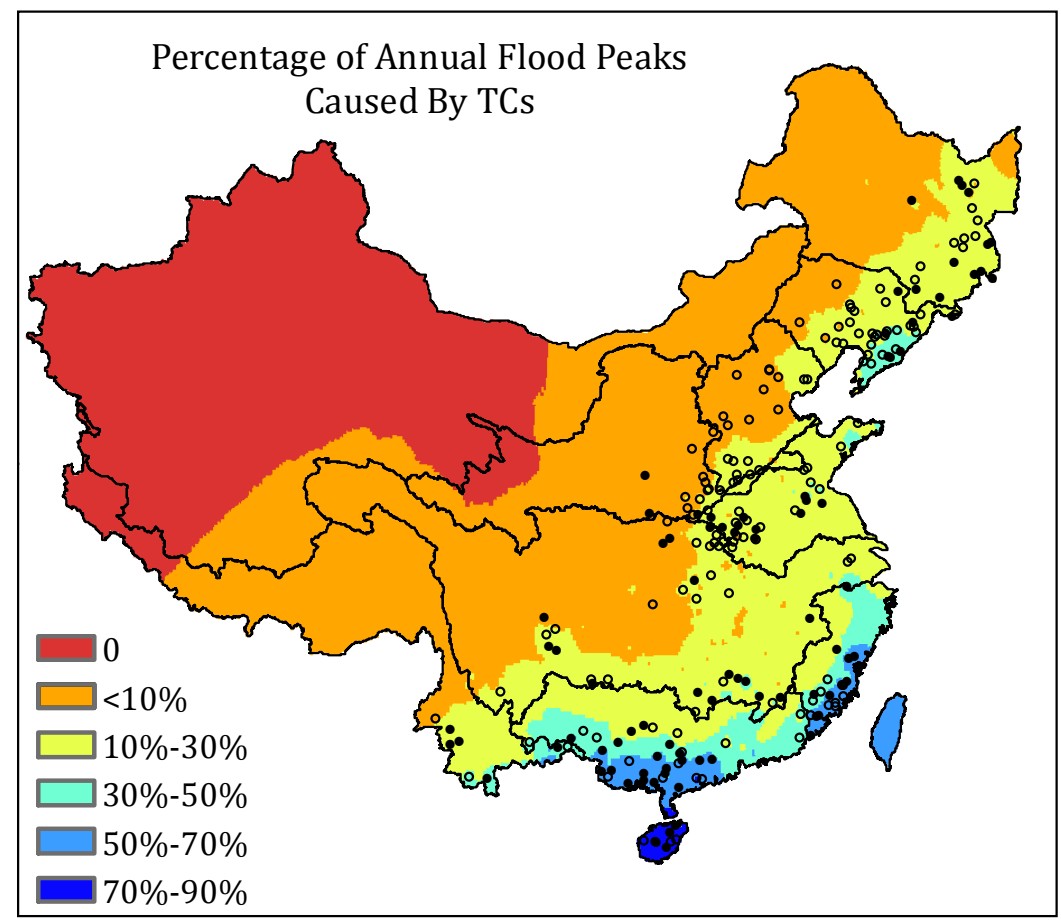

**Figure 10.** Percentage of annual flood peaks that are caused by tropical cyclones. The black dots and circles represent the stations with record floods caused by tropical cyclones. The black dots further highlight stations with stationary time series of annual flood peaks.

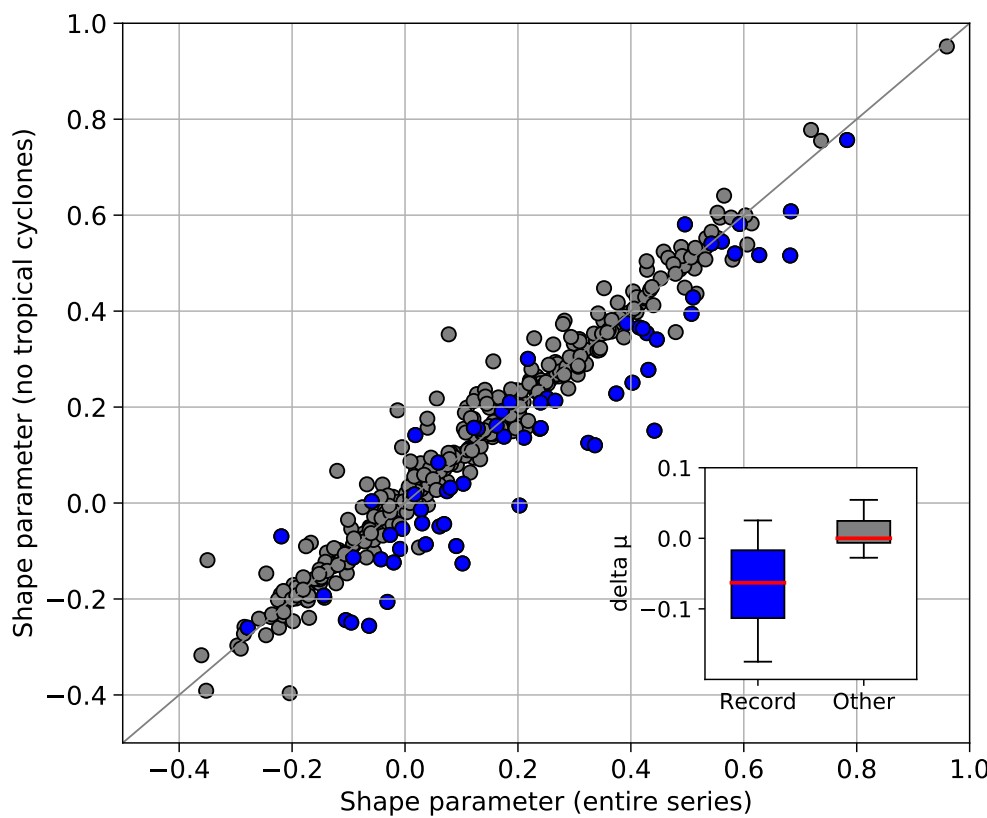

**Figure 11.** Scatterplot of the shape parameters for the entire series versus the series with annual flood peaks caused by tropical cyclones removed. Blue dots highlight the stations with record floods that are caused by tropical cyclones (see Figure 1 for locations). The insert boxplot shows the differences of shape parameter (series with TC flood peaks removed minus the entire series) for stations with (blue) and without (grey) TC-induced record floods.

**Figure 12.** Tropical cyclones that produced more than 100 annual flood peaks (blue dots) over China. Red dots highlight that the annual flood peak is also the record flood of the station. Dark black line shows tropical cyclone track. See Table 1 for more details.