# Peer review of "On the Flood Peak Distributions over China"

_Hydrology and Earth System Sciences, 2019_

## Short Comment (SC1) · 17 Jul 2019

I do think this study is interesting, particularly, some findings were based on good dataset. However, I do find serious issues to be address before potential acceptance. (1) What are the scientific issues or scientific assumptions to be addressed? What are the objectives of this current study? (2) As for detection of change points, results by one statistical method are not certain with considerable uncertainty. This issue was well addressed by Zhang et al. (2009), i.e. Qiang Zhang, Chong-Yu Xu, Yongqin David Chen, Jianmin Jiang, 2009. Abrupt behaviors of the streamflow of the Pearl River basin and implications for hydrological alterations across the Pearl River Delta, China. Journal of Hydrology, 377(3), 274-283. By the way, the assumption that only one change point can be observed in one streamflow series is not practically and theoretically correct. It is definitely wrong! (3) The authors tried to relate GEV parameters to tropical cyclones and fit GEV model to stationary series. This kind of analysis is totally wrong. I

suggest stationary GEV model for stationary flood peak series, but nonstationary GEV, i.e. GEV with time-varying parameters, for nonstationary flood peak series. Moreover, flood peak process is not the result of tropical cyclone only, but most flood processes are by extreme precipitation. Therefore, I suggest association of peak flood flows to extreme precipitation but not tropical cyclones. Just as said, some extreme precipitations are by tropical cyclones. (4) Peak flood flows are heavily influenced by water reservoirs. Actually, it was done by Zhang et al. (2015), i.e. Qiang Zhang, Xihui Gu, Vijay P. Singh, Chong-Yu Xu, Dongdong Kong, Mingzhong Xiao, Xiaohong Chen, 2015. Homogenization of precipitation and flow regimes across China: changing properties, causes and implications. Journal of Hydrology, 530, 462-475. Relevant case studies can be found as: Qiang Zhang, Vijay P. Singh, Chong-Yu Xu and Xiaohong Chen, 2013. Abrupt behaviors of streamflow and sediment load variations of the Yangtze River basin, China. Hydrological Processes, 27(3), 444-452. Impoundment effects of water reservoirs on flood processes cannot be ignored in this study.

---

## Referee Comment (RC1) · Hong Do (Referee) · 26 Aug 2019

**Long Yang, Lachun Wang, Xiang Li, and Jie Gao, "On the Flood Peak Distributions over China",
submitted to Hydrology and Earth System Sciences (hess-2019-322).**

This paper presents an investigation of the characteristics of annual maximum streamflow using a
comprehensive database of more than 1000 stream gauges across China. The authors also related
flood occurrences and associated changes to catchment regulations and other climatic drivers with a
focus on tropical storms. The authors' effort in collating and analysing streamflow data from
different sources is a significant contribution to large-scale hydrology. I'm particularly impressed by
the database being used, and found the paper has the potential to make significant contribution to
the field, at both regional and global scales.

To make the contributions of this study really come through, the authors must address several major
issues. Most of these issues belongs to the structure and content, but there are also some
methodological issues. I outlined these major issues in "General Comments" section, followed by
more specific comments to further clarify my concerns. I'm recommending a major revision to
provide the authors sufficient time to thoroughly revise the manuscript.

**NOT CONTENT RELATED**

Perhaps this is an editorial decision but I found the bottom-left panel of Figure 1 (the marine region
southern of China) irrelevant to the scientific content of this paper, as there was no gauge located in
that area. Please also note that there is political tension around this region due to ongoing territorial
disputes involved several countries (see
https://en.wikipedia.org/wiki/Territorial_disputes_in_the_South_China_Sea). As political science is
not the focus of this study, I strongly recommend omitting this feature from the map to avoid
unnecessary debates.

**GENERAL COMMENTS**

**1. Scientific writing issues**

**The nature of an analogous study**

This paper shares a significant similarity to that of Villarini and Smith (2010) in terms of manuscript
structure, narrative, methodology and content. As indicated at the beginning of my review, the
findings are still significant contributions to the literature as this study used an unprecedented
database for China and have improved the state-of-understanding of flood hazard at the regional
and global scale. Nevertheless, I strongly recommends the authors to revise the manuscript
substantially (i.e. making the manuscript more independent to its counterpart for the eastern of US)
as the presented study has more potential than a simple incremental research.

I would also consider it appropriate to highlight the link between this analysis to similar works across
the globe to place this contribution in a broader context of current assessments and datasets being
used worldwide to assess flood peak characteristics. There are some studies for other regions
mentioned in the introduction, but they have not been discussed systematically to highlight the
significance of a China-wide investigation. Considering the quality/magnitude of the database and
analyses presented, I consider the study's most significant contribution (which unfortunately was
not adequately highlighted in the current manuscript) is to complement the limited understanding of
flood characteristics at the global scale (e.g. changes in flood magnitude and timing). Specifically,
recent observation-based findings on flood hazard magnitude and timing (Do et al., 2019;Do et al.,
2017;Hodgkins et al., 2017;Burn and Whitfield, 2018;Mangini et al., 2018;Hall and Blöschl,
2018;Slater and Villarini, 2016) have not provided much information for Asia (note that China

contributes a significant share to Asia land area). The key barrier is the insufficient streamflow records presented with the most comprehensive global datasets to-date such as GRDC or GSIM (Do et al., 2018a;Gudmundsson et al., 2018a), and thus large-scale hydrologists do not possess a representative sample of streamflow observation (especially for Asia). Current understanding of global flood hazards may therefore be biased toward Europe and North America, making studies similar to the present study very demanding.

If the authors prefer to stay on the current objective (i.e. conducting an analogous study of a previous study), I want to see this objective introduced more prominent (e.g. explain why a nearly identical study was conducted) as it is currently mentioned very vaguely in the introduction (page 2, lines 35-37). This potentially makes a notion that the authors did not give Villarini and Smith (2010) the merits it deserves. The authors should also focus on clarifying the technical aspects that were identically repeated and the ones that were done differently (beside different datasets and study areas) – the rationale of these choices should also be discussed. In addition, "distributions" in Villarini and Smith (2010) refer to several terminologies (e.g. spatial, temporal and statistical distributions) and the rationale of using this word was presented adequately in the introduction of that paper. The authors must rethink the key motivation of their study to clarify which "distributions" are the focus of their research and justify why a specific characteristic of flood peaks is chosen (similar to the link between Typhoon Nina and timing of flood peaks currently presented). The loose ties between the "distributions" and the objectives currently made the paper read fragmented and confused.

If the authors decide to follow another approach, the study may excluded some analyses (e.g. GEV distribution) to focus more on an aspect of flood hazards (e.g. trends in flood magnitude and timing) and relate the paper findings to previous observation-based studies. As indicated above, with this unprecedented dataset, I believe there are several rooms for analyses and discussions beyond the current strategy (i.e. an analogous study of Villarini and Smith (2010)). There are opportunities for deeper and more critical discussions included in the "Specific comments" section that are applicable if the authors choose this direction.

**Abstract:**

This section should be revised significantly to provide a clear synthesis of the objectives, the methods being used and the main findings. The opening sentence reiterates the ambiguous terminology presented in the title (i.e. "flood peak distributions") without sufficient clarifications in the rest of the abstract. There are several statistical analyses presented with little "hints" about the role of each assessment as well as the links between the results and specific objectives. This shortcoming has led to the notion that the paper is a collection of unrelated analyses (which is not the case when reading further to the main text).

**Introduction**

After reading through the introduction twice, I'm still unclear about "what are the research questions being introduced?", and "what are the key contributions of this study?" The confusions partly come from the loose link between the ideas being presented and the choice of terminologies/wordings that sometimes overcomplicates the ideas. This section must be revised to provide a more synthesized literature review and simplify the ideas where possible. Please also refer to my previous comments for some thoughts about improving the introduction. The authors may find some of the "specific comments" useful as well.

**Results and Discussions**

This section was too "report focused" with very loose connection between individual sub-sections. This led to a notion that the "Results and Discussion" is a compilation of four separate studies. I strongly recommend the authors to remedy this issue and better explain the contribution of each sub-section to the overarching aims. One possible way is to have an "introduction sentence/paragraph" at the beginning of each sub-section to clarify how the subsequent discussion contributes to a better understanding of "flood peak distributions for China".

**Conclusions**

This section reads more "Summary" than "Conclusion". It would be more useful if the authors be more concise in summarizing the key findings and then focus on highlighting (1) what was the contribution of this study to the state-of-understanding for flood hazard (China-wide and global scale) and (2) the fact that collected data set is unprecedented and has much more potentials beyond this study (e.g. adding values to the literature of global scale hydrology). It would beneficial to also comments on the benefit, challenges and potential of making China streamflow data becomes more FAIR (Findable, Accessible, Interoperable and Reusable; see Wilkinson et al., 2016). Some potential options toward this ambitious goal is to publish metadata and indices (Do et al., 2018b;Gudmundsson et al., 2018b) and even include hydrological variable time series (Addor et al., 2017).

**2. Methodological issues**

There are some data aspects that should be clarified in Section 2 (or presented in supplementary) as this is the first study to use such dataset, which may be unfamiliar to other hydrologists. Some recommended clarifications:

- Data quality statements. The definition of "continuous records of at least 50 years" (e.g. non-missing data consecutively for 50 years) and "Strict quality control procedures are implemented to ensure consistency and accuracy of the records". These aspects are very important on the credibility of detectable trends.
- Filtering criteria related to missing data. Please note that annual value of maximum are very sensitive to missing data. Previous studies usually applied a threshold of number of missing data points per year to assign "N/A" value for a specific year (e.g. station A has 30 missing data points in 1950 so N/A value was assigned for that particular year). If all gauges have zero missing data-point, this feature should be highlighted to show the exceptionally good quality of this dataset.
- A graph presents the number of stations with available data over time may be useful. This will also provide the an understanding of the uncertainty related to step-change analysis. Figure 2 of Do et al. (2017) is an example for this type of plot.
- Figure 1 should show the location of "discarded stations" prior to further analyses. It seems to me that the station density reduced significantly in other figures. If all stations showed in Figure 1 were used, please note in the caption. If there are stations removed, the number of stations used/removed for each subsequent analysis should also be mentioned.
- Additional maps should be added to Figure 1 to show (i) data length, (ii) the beginning and (iii) end of records for each station to complement the subsequent analysis of change-point. For instance, one hypothesized reason for change point detected mostly over 1980-2010 was most stations have streamflow data available for only 1970s onward.

Trends analysis: as a China-wide investigation is one of the key motivations of this study, I'm recommending the use of "reference period". The authors may conduct the analysis for only one

period (e.g. all stations were assessed for 1969-2019) or different period (e.g. 1980-2010, 1950-2019, and 1900-2019). This is particular important to remedy data limitation, which could dismiss the usefulness of the detected trends wherever too short records was used to assess trend in floods over a long period (e.g. using 50 years of data to infer changes in floods for a 150-year period).

For Mann-Kendall test: is there any criterion for data length to conduct the analysis (e.g. at least ten data points)? This is relevant for the test conducted for before-step-changes and after-step-changes time series (there may be insufficient data points). I noted that the figures 5b and 5c have different numbers of stations, potentially due to insufficient data points for a specific sub-group?

For flood timing, it is unclear the motivation for assessing trends in this flood index. In addition, it is more useful to use an analysis where the magnitude of changes is visible. The reason is even when changes are significant, it is not practically meaningful if flood timing shifts only a small time-window (e.g. one day over 100 years). Theil-Sen slope estimator is a useful statistical technique for this type of analysis (Blöschl et al., 2017).

**SPECIFIC COMMENTS**

Title: "Flood Peak Distributions over China" made the notion that "spatial distribution" is the key feature being analysed (which is not the case).

Line 2: "flood peak distributions across China" in the abstract also made the notion that "spatial distribution" is the key feature being analysed.

Line 16: the research objectives were "to provide improved understandings on the nature of upper tails of flood peaks and innovative methods for flood frequency analysis in a changing environment". This statement has two issues: (1) it is unclear what is "the nature of upper tails of flood peaks"? (Any alternative for "upper tails of flood peaks", which was used quite often across the manuscript?), and (2) "provide innovative methods for flood frequency analysis" indicates the development of a new method, which is not the focus of this paper. The goal should be rewritten.

Line 17: It is unclear how the four presented themes linked to the two objectives.

Line 26: "staionarity" should be "stationarity".

Line 35: "highlight possible factors that induce the changes" should be clarified: some of potential factors and why did they are chosen?

Line 36: it is unclear to me what "dominant modes of violation for the stationarity assumption" is. Would there be a more simple way to explain it?

Line 37: the final sentence seems out of place. I was expecting a clarification of the "dominant modes..." mentioned it the previous sentence, or justification of why the authors followed Villarini and Smith (2010) rather than Hodgkins et al. (2019).

Line 46: it is unclear what "space-time rainfall organizations" means.

Line 48: I don't think the sub-sequent sections discussed anything related about "the necessity of improved procedures for regional flood frequency analysis with spatial heterogeneity in flood hydro-climatology considered". Please revise this statement or extend the discussion.

Lines 52: please note that "seasonality" may refer to more than the timing of floods (Villarini, 2016).

Lines 57-62: please focus more on introducing the mechanisms generating floods across China.

Line 63: it is unclear what "monsoon-related systems" means.

Lines 64-67: these two sentences read out of place.

Line 67-68: this section is repetitive. I'm recommending the authors to link this paragraph to the subsequent one (starting at line 70) as timing of tropical cyclones were then related to flood timing.

Lines 70-92: I found the detail introduction of Typhoon Nina is too disruptive – and a distraction. This is also a reason for confusion about the key research question of this research. If the key motivation was "to examine the impact of tropical cyclone on the upper tail properties of flood peak distribution over China" (line 85; and I think this is a great research question itself), the structure of the manuscript need to be revised to reflect this main research objective. The current manuscript presents the link between floods and tropical storms at the very end (section 4.4), after a very long discussions for other factors (e.g. step-changes, seasonality, GEV...) with almost no "reminder" for readers. At the time I got to section 4.4, I almost forgot the motivation to analyse "Tropical cyclones and upper tail properties" in this study.

Line 88: I'm not sure why "Results presented in this study can promote a predictive understanding of flood hazards associated with landfalling tropical cyclones".

Line 121: please explain the rationale of applying Mann-Kendall test for two sub-groups and the contribution of this analysis to the main objective.

Line 124: "significance level of 5%" is for one-tail or two-tail?

Line 125: please justify why circular statistic, a common approach for flood timing investigations (Villarini, 2016;Blöschl et al., 2017;Hall and Blöschl, 2018), was not used in this study to prevent the issues where two flood peaks occurring on calendar day 1 and calendar 365 only have one day difference.

Lines 136-140: this is repetitive (has been mentioned in introduction). The authors should clarify about how to examine the dependence of GEV parameters on drainage area.

Line 145: is there any reference for this choice (i.e. 500km and two weeks).

Lines 149-155: these information seems irrelevant as only the circulation centre location was used. In case the sub-sequent analyses will divide tropical storms into sub-categories (ET/TS), the authors should clarify this technical aspect.

Line 160: "The majority of stations tend to show smaller values". Please clarify what is "the majority" and how "smaller" the values are (perhaps in %).

Line 169: "We are able to relate some of the changes in annual flood peaks series to intentional human activities". Please clarify the procedure used to identify these relationships (e.g. metadata inspection?) and what is the magnitude of "some" (e.g. 5% of stations with significant step-change detected?).

Figure 3: Please also show the locations of gauges with insignificant results. The geographical location of individual stations analysed in Figure 4 should also be highlighted (e.g. starred symbol).

Line 197: this is somewhat expected as changes in climate variables occur quite gradually in general. Previous studies generally link abrupt changes to human interventions rather than natural climate drivers.

Line 203: please be careful with your conclusion that "Abrupt change rather than slowly varying trend is a common mode of the violation of the stationarity assumption for the annual flood peak series over China" as both modes of non-stationarity can present at the same station. For instance, "naturalized streamflow records" (e.g. the difference in means between two before/after step-change time series were removed) may reveals gradual change, which is more relevant to climate changes and variabilities. As a result, assessing linear trends over only stations that did not exhibit significant step-change is not sufficient to support this statement.

Figure 5b and Figure 5c: please clarify why these two figures have different number of data points. If it is due to statistical insignificance, please also show the location of stations that did not exhibit significant linear trends.

Line 227: Please note that the findings do not support the statement that "external climate factors (i.e., extreme rainfall), and changes in soil moisture on flood hydrology" leads to flood stationarity.

Lines 232-235: it is unclear what "state-of-art process-based approaches" and "statistical modelling approaches" are – please clarify. Please also make it clearer why these approaches important to "flood frequency analyses across China" (i.e. how could these approaches address the non-stationarities in flood frequency).

Section 4.2: maps of the average flood timing and associated concentration (Villarini, 2016) would be a nice addition.

Lines 239: "the first peaks"? Considering the distribution of the floods timing (Figure 6), I thought there is only one peak per group?

Lines 240-241: please provide reference.

Line 252: please clarify how "tropical cyclone floods" defined (this should be presented in methodology).

Line 260: forward-reference is not recommended.

Lines 265-280: without sufficient evidence of "how many days flood peaks have shifted", it is hard to justify these statements.

Line 271: "Villarini (2016) found …" sounds out of place.

Lines 273-278: these statements read contradicting to each other.

Line 287: please clarify why only these stations used (should explain in the methodology).

Line 312: "contrasting space-time organizations" is unclear. Please clarify.

Line 343: the presented results only show the impacts of tropical cyclone on flood occurrence rather than "flood peak distributions".

Figure 12: please plot also the stations within 500-km distant and the annual maximum streamflow does not coincide with the occurrence of the selected tropical cyclones (i.e. flood timing is outside the two weeks threshold) and extend the discussions appropriately (e.g. the proportions of stations influenced by tropical storms).

**References**

Addor, N., Newman, A. J., Mizukami, N., and Clark, M. P.: The CAMELS data set: catchment attributes and meteorology for large-sample studies, Hydrol. Earth Syst. Sci. Discuss., 2017, 1-31, 10.5194/hess-2017-169, 2017.

Blöschl, G., Hall, J., Parajka, J., Perdigão, R. A. P., Merz, B., Arheimer, B., Aronica, G. T., Bilibashi, A., Bonacci, O., Borga, M., Čanjevac, I., Castellarin, A., Chirico, G. B., Claps, P., Fiala, K., Frolova, N., Gorbachova, L., Gül, A., Hannaford, J., Harrigan, S., Kireeva, M., Kiss, A., Kjeldsen, T. R., Kohnová, S., Koskela, J. J., Ledvinka, O., Macdonald, N., Mavrova-Guirguinova, M., Mediero, L., Merz, R., Molnar, P., Montanari, A., Murphy, C., Osuch, M., Ovcharuk, V., Radevski, I., Rogger, M., Salinas, J. L., Sauquet, E., Šraj, M., Szolgay, J., Viglione, A., Volpi, E., Wilson, D., Zaimi, K., and Živković, N.: Changing climate shifts timing of European floods, Science, 357, 588, 2017.

Burn, D. H., and Whitfield, P. H.: Changes in flood events inferred from centennial length streamflow data records, Advances in Water Resources, 121, 333-349, https://doi.org/10.1016/j.advwatres.2018.08.017, 2018.

Do, H. X., Westra, S., and Michael, L.: A global-scale investigation of trends in annual maximum streamflow, Journal of Hydrology, 10.1016/j.jhydrol.2017.06.015, 2017.

Do, H. X., Gudmundsson, L., Leonard, M., and Westra, S.: The Global Streamflow Indices and Metadata Archive (GSIM) – Part 1: The production of a daily streamflow archive and metadata, Earth Syst. Sci. Data, 10, 765-785, 10.5194/essd-10-765-2018, 2018a.

Do, H. X., Gudmundsson, L., Leonard, M., and Westra, S.: The Global Streamflow Indices and Metadata Archive - Part 1: Station catalog and Catchment boundary, in, PANGAEA, 2018b.

Do, H. X., Zhao, F., Westra, S., Leonard, M., Gudmundsson, L., Chang, J., Ciais, P., Gerten, D., Gosling, S. N., Schmied, H. M., Stacke, T., Stanislas, B. J. E., and Wada, Y.: Historical and future changes in global flood magnitude – evidence from a model-observation investigation, Hydrol. Earth Syst. Sci. Discuss., 2019, 1-31, 10.5194/hess-2019-388, 2019.

Gudmundsson, L., Do, H. X., Leonard, M., and Westra, S.: The Global Streamflow Indices and Metadata Archive (GSIM) – Part 2: Quality control, time-series indices and homogeneity assessment, Earth Syst. Sci. Data, 10, 787-804, 10.5194/essd-10-787-2018, 2018a.

Gudmundsson, L., Do, H. X., Leonard, M., and Westra, S.: The Global Streamflow Indices and Metadata Archive (GSIM) - Part 2: Time Series Indices and Homogeneity Assessment, in, PANGAEA, 2018b.

Hall, J., and Blöschl, G.: Spatial patterns and characteristics of flood seasonality in Europe, Hydrol. Earth Syst. Sci., 22, 3883-3901, 10.5194/hess-22-3883-2018, 2018.

Hodgkins, G. A., Whitfield, P. H., Burn, D. H., Hannaford, J., Renard, B., Stahl, K., Fleig, A. K., Madsen, H., Mediero, L., Korhonen, J., Murphy, C., and Wilson, D.: Climate-driven variability in the occurrence of major floods across North America and Europe, Journal of Hydrology, 552, 704-717, http://dx.doi.org/10.1016/j.jhydrol.2017.07.027, 2017.

Hodgkins, G. A., Dudley, R. W., Archfield, S. A., and Renard, B.: Effects of climate, regulation, and urbanization on historical flood trends in the United States, Journal of Hydrology, 573, 697-709, https://doi.org/10.1016/j.jhydrol.2019.03.102, 2019.

Mangini, W., Viglione, A., Hall, J., Hundecha, Y., Ceola, S., Montanari, A., Rogger, M., Salinas, J. L., Borzì, I., and Parajka, J.: Detection of trends in magnitude and frequency of flood peaks across Europe, Hydrological Sciences Journal, 63, 493-512, 10.1080/02626667.2018.1444766, 2018.

Slater, L. J., and Villarini, G.: Recent trends in U.S. flood risk, Geophysical Research Letters, 43, 12,428-412,436, 10.1002/2016GL071199, 2016.

Villarini, G., and Smith, J. A.: Flood peak distributions for the eastern United States, 46, 10.1029/2009wr008395, 2010.

Villarini, G.: On the seasonality of flooding across the continental United States, Advances in Water Resources, 87, 80-91, https://doi.org/10.1016/j.advwatres.2015.11.009, 2016.

Wilkinson, M. D., Dumontier, M., Aalbersberg, I. J., Appleton, G., Axton, M., Baak, A., Blomberg, N., Boiten, J.-W., da Silva Santos, L. B., Bourne, P. E., Bouwman, J., Brookes, A. J., Clark, T., Crosas, M.,

Dillo, I., Dumon, O., Edmunds, S., Evelo, C. T., Finkers, R., Gonzalez-Beltran, A., Gray, A. J. G., Groth, P., Goble, C., Grethe, J. S., Heringa, J., 't Hoen, P. A. C., Hooft, R., Kuhn, T., Kok, R., Kok, J., Lusher, S. J., Martone, M. E., Mons, A., Packer, A. L., Persson, B., Rocca-Serra, P., Roos, M., van Schaik, R., Sansone, S.-A., Schultes, E., Sengstag, T., Slater, T., Strawn, G., Swertz, M. A., Thompson, M., van der Lei, J., van Mulligen, E., Velterop, J., Waagmeester, A., Wittenburg, P., Wolstencroft, K., Zhao, J., and Mons, B.: The FAIR Guiding Principles for scientific data management and stewardship, Scientific Data, 3, 160018, 10.1038/sdata.2016.18, 2016.

---

## Referee Comment (RC2) · Anonymous Referee #2 · 30 Aug 2019

The authors present an extensive study on exploring flood peak distributions in terms of stationarity, seasonality, scaling property, spatial heterogeneity and the effects of land-falling tropical cyclones across the China. Their results are mainly based on several statistical methods (e.g. Pettitt's test, Mann-Kendall test, GEV distribution, etc.) using annual maximum peak discharge from over 1000 gages with a record of at least 50 years. To my knowledge, their study is the first to analyze the characteristics of annual maximum flood peaks on a nation-wide scale in China, highlighting the importance of both streamflow dataset and treating them not only as numerical values but as real world physical events.

Overall, this study is organized and well-written. However, I do have some major concerns, which are listed below, about the structures and certain elements of the study that need to be addressed before the work can be considered for publication. Typographical errors (e.g. line 255, "Tainhang Mountains" should be "Taihang Mountains")

[Figure]

in the paper can be addressed in the second round of review because some of them might be removed during author's revision.

Specific comments: 1. The main objective or scientific question of this study are not adequately addressed that causes each part of analysis does not connect logically. Authors have applied several statistical approaches with peak flood data but the explanation of the necessity and connection of each test need to be more explicitly addressed. I understand that similar study was done by Villarini and Smith (2010) US but the authors need to demonstrate why Villarini and Smith's analysis are also necessary in this study. For example, why do authors decide to explore the role of tropical cyclone rather than snowmelt in characterizing the upper tail of flood peaks?

2. Although authors conclude that abrupt changes in flood magnitude and seasonality are mainly due to anthropogenic influence, I would suggest them to investigate (or even focus on) climate induced nonstationarity. The anthropogenic induced changes imply stationarity after changing point (e.g. the built of reservoir, urbanization), which is not valid in this study, as shown in Figure 5, highlighting the existence of climate driven changes. The author's explanation for Figure 5c is vague, such as these negative trend might be due to soil conservation or decreasing rainfall intensity. That is why I suggest authors to separate climate-induced changes from anthropogenic-induced one when discussing the violation of stationarity in China. The other reason is that climate-induced changes around the world has already been unearthed, and author's contribution might provide such understandings in China. For example, Blöschl et al. (2017, 2019) found changing climate derives changes in flood seasonality and magnitude over Europe using streamflow data. Blöschl, Günter, et al. "Changing climate shifts timing of European floods." Science 357.6351 (2017): 588-590. Blöschl, Günter, et al. " Changing climate both increases and decreases European river floods." Nature 1476. 4687 (2019): https://doi.org/10.1038/s41586-019-1495-6.

A minor relevant suggestion is to modify the label in Figure 7b, indicating clearly which trend represents flood seasonality shifting to earlier or later time in a year.

Interactive
comment

3. Despite the significance of flood seasonality as authors addressed, the section 4.2 of this paper and other relevant parts failed to present how flood seasonality is distributed across the country and how to link them to flood generating mechanisms. I would suggest authors to add two maps: average seasonality of all maximum annual floods and of only the three (or two) biggest floods for all gages across the country. These two maps can indicate the regional patterns of flood seasonality and how flood processes may change as one moves from moderate to extreme floods. They can also help authors demonstrate some of their arguments, such as line 240-242, "Frequent occurrence of annual..." Without a map, it is very difficult for reader to envision how flood seasonality is distributed across the country.

4. An interesting finding in this paper is that tropical cyclones (monsoon-controlled storms) plays key role in determining upper tail of flood peaks in northern (southern) China (line 252-264), but needs to be well defended. Figure 6 only shows that flooding happens more frequently around June (July) in southern (norther) China, but fails to present their severity (magnitude). For instance, is it possible the most frequent floods in June over southern China associate with low or moderate severity (i.e. magnitude)?

5. Line 54-56 ("Annual flood peaks resulted..."): I suggest being more careful in your wording here. I would respectfully disagree with authors that conventional flood frequency analysis (FFA) requires a homogenous flood population with respect to flood-generating mechanisms. Instead, a flood series at any gage is a mixture of different flood-generating processes but they are just one sample not the population. Conventional FFA assumes peak discharge measured at a gage over a finite period is a sample from the population of all possible floods (representing different flood generating mechanisms) during an undefinable length of time.

6. The authors should provide a brief description of the quality control procedures in section two rather than just state they have done so. This procedure is important in this study since datasets are from different sources and inaccessible to public. For instance, what is the time interval for the "instantaneous" peak discharge and are they
the same value for all gages? If the instantaneous peak discharge data were from US Geological Survey (USGS), I would expect they are all in the 15-minute interval but have no clue here.

---

## Author Comment (AC1) · 16 Sep 2019

**Responses to short comments**

*(1) What are the scientific issues or scientific assumptions to be addressed? What are the objectives of this current study?*

**Response:** We thank the reviewer for this critique. We substantially reconstruct Abstract and Introduction sections to highlight the objectives and scientific issues of the present study in the revised manuscript. In short, we expect to provide improved characterizations of flood hazard over China from both statistical and physical perspectives, and contribute to improved understandings of flood hydrology and hydroclimatology under a changing environment. Our analysis is carried out by centering on five proposed questions (see Line 88-93 of the revised manuscript). Our analysis is based on an unprecedented dataset that can significantly advance flood science at the global scale (as also highlighted by the other two reviewers). Thanks!

*(2) As for detection of change points, results by one statistical method are not certain with considerable uncertainty. This issue was well addressed by Zhang et al. (2009), i.e. Qiang Zhang, Chong-Yu Xu, Yongqin David Chen, Jianmin Jiang, 2009. Abrupt behaviors of the streamflow of the Pearl River basin and implications for hydrological alterations across the Pearl River Delta, China. Journal of Hydrology, 377(3), 274-283.*

**Response:** We thank the reviewer for this critique. Some other change-point detection approaches have been applied, but only show negligible deviations from those by Pettitt's test. There are some variations for specific stations, but the years of change points occur in 1980s and clustered in central China and northern China. We below show comparisons of change points in mean based on Pettitt's test and the one proposed by Matteson and James (2014). Our conclusions remain unchanged using different methods. We clarify this in the revised manuscript. Thanks!

[Figure]

Figure R1. Comparisons between results using different change-point detection methods. (a) Pettitt's test, (b) The approach proposed by Matteson and James (2014).

*(3) By the way, the assumption that only one change point can be observed in one streamflow series is not*

*practically and theoretically correct. It is definitely wrong!*

**Response:** We agree with the reviewer that multiple change points can exist in a flood series. However, we are particularly interested in understanding the dominant mode of nonstationarity in flood series across China, i.e., abrupt change vs. slowly varying trend, rather than locating every single possible change point in the series. As we have mentioned in the manuscript, "we assume the existence of only a single change point in mean for each flood peak series in this study, to avoid dividing the series into too many segments". In addition, the assumption of one single change point is also frequently adopted in previous studies (e.g., Villarini 2009; 2010; 2012). We add references in the revised manuscript. Thanks!

*(4) The authors tried to relate GEV parameters to tropical cyclones and fit GEV model to stationary series. This kind of analysis is totally wrong. I suggest stationary GEV model for stationary flood peak series, but nonstationary GEV, i.e. GEV with time-varying parameters, for nonstationary flood peak series. Moreover, flood peak process is not the result of tropical cyclone only, but most flood processes are by extreme precipitation. Therefore, I suggest association of peak flood flows to extreme precipitation but not tropical cyclones. Just as said, some extreme precipitations are by tropical cyclones.*

**Response:** We believe the reviewer has accidentally misunderstood some of the analysis conducted for tropical cyclones and their roles in determining the upper-tail properties of flood peak distributions. GEV analysis (with time-independent parameters) is only carried out for flood series that are stationary (i.e., demonstrating no abrupt changes or monotonic trends). We examine the role of tropical cyclones in determining the upper tails of flood peak distributions mainly through examining changes in the GEV shape parameters after removing tropical cyclone-induced flood peaks from the entire flood series (as shown in Figure 10, the revised manuscript). Nonstationary GEV modeling based on time-dependent parameters is not the focus of our present study.

A distinct feature of tropical cyclones is the spiral rainbands that typically extend from the eye wall towards several hundred or even tens of hundreds of kilometers away from the center of circulation. Climatological analysis based on satellite rainfall products shows that most extreme rainfall is distributed within 500 km around the circulation center (e.g., Rios Gaona et al., 2018). This is the reason why we choose 500 km as the spatial threshold to associate flood peaks with a tropical cyclone, with the essence being actually association of flood peaks with extreme rainfall induced by tropical cyclones. Thanks all the same!

*(5) Peak flood flows are heavily influenced by water reservoirs. Actually, it was done by Zhang et al. (2015), i.e. Qiang Zhang, Xihui Gu, Vijay P. Singh, Chong-Yu Xu, Dongdong Kong, Mingzhong Xiao, Xiaohong Chen, 2015. Homogenization of precipitation and flow regimes across China: changing properties, causes and implications. Journal of Hydrology, 530, 462-475. Relevant case studies can be found as: Qiang Zhang, Vijay P. Singh, Chong-Yu Xu and Xiaohong Chen, 2013. Abrupt behaviors of streamflow and sediment load variations of the Yangtze River basin, China. Hydrological Processes, 27(3), 444-452. Impoundment effects of water reservoirs on flood processes cannot be ignored in this study.*

**Response:** We agree with the reviewer that reservoirs play an important role in flood peak magnitudes. We highlight this by showing a flood series in the upper Yellow River basin in Figure 4. However, we also note

that the effects of reservoirs on flood peak may depend on some other factors, due to contrasting findings of previous studies (e.g., Smith et al., 2010; Barros et al., 2014). We add the reference Zhang et al. (2015) in the revised manuscript. We specifically discuss the influence of reservoirs in Line 198-202 of the revised manuscript. Thanks!

*References:*

[revised manuscript text omitted]

---

## Author Comment (AC2) · 16 Sep 2019

**Responses to Reviewer 1**

*(1) This paper presents an investigation of the characteristics of annual maximum streamflow using a comprehensive database of more than 1000 stream gauges across China. The authors also related flood occurrences and associated changes to catchment regulations and other climatic drivers with a focus on tropical storms. The authors' effort in collating and analyzing streamflow data from different sources is a significant contribution to large-scale hydrology. I'm particularly impressed by the database being used, and found the paper has the potential to make significant contribution to the field, at both regional and global scales. To make the contributions of this study really come through, the authors must address several major issues. Most of these issues belongs to the structure and content, but there are also some methodological issues. I outlined these major issues in "General Comments" section, followed by more specific comments to further clarify my concerns. I'm recommending a major revision to provide the authors sufficient time to thoroughly revise the manuscript.*

**_Response_:** We really appreciate the reviewer's efforts and time on our manuscript. We revise the manuscript substantially based on all the comments, and make a point-by-point response below. The reviewer's comments are enumerated. Our replies to each comment start with "**_Response_**". Thanks!

*(2) Perhaps this is an editorial decision but I found the bottom-left panel of Figure 1 (the marine region southern of China) irrelevant to the scientific content of this paper, as there was no gauge located in that area. Please also note that there is political tension around this region due to ongoing territorial disputes involved several countries (see https://en.wikipedia.org/wiki/Territorial_disputes_in_the_South_China_Sea). As political science is not the focus of this study, I strongly recommend omitting this feature from the map to avoid unnecessary debates.*

**_Response_:** We prefer to keep the bottom-left panel as where it is. As the reviewer also note that it is an editorial decision, we will follow the editor's advice on this particular point. Thanks!

*(3) The nature of an analogous study: this paper shares a significant similarity to that of Villarini and Smith (2010) in terms of manuscript structure, narrative, methodology and content. As indicated at the beginning of my review, the findings are still significant contributions to the literature as this study used an unprecedented database for China and have improved the state-of-understanding of flood hazard at the regional and global scale. Nevertheless, I strongly recommend the authors to revise the manuscript substantially (i.e. making the manuscript more independent to its counterpart for the eastern of US) as the presented study has more potential than a simple incremental research.*

**_Response_:** We thank the reviewer for this comment. We indeed adopt similar methods as used in Villarini and Smith (2010), but target at different questions. As we have emphasized in the revised manuscript that the ultimate goal of the present study is to provide better characterization of flood hazard over China from both statistical and physical perspective. We appreciate that the reviewer realizes this point as well. The goal leads to a different structure of the paper. For instance, a major part of the present study is to understand the impacts of landfalling tropical cyclones on the upper-tail properties of flood peaks across China and the key

features of tropical cyclones that lead to most extreme floods. These contents together with the research goal is not what Villarini and Smith pursued in their paper. We reconstruct the Abstract and Introduction in the revised manuscript to make this clear. Thanks!

*(4) I would also consider it appropriate to highlight the link between this analysis to similar works across the globe to place this contribution in a broader context of current assessments and datasets being used worldwide to assess flood peak characteristics. There are some studies for other regions mentioned in the introduction, but they have not been discussed systematically to highlight the significance of a China-wide investigation. Considering the quality/magnitude of the database and analyses presented, I consider the study's most significant contribution (which unfortunately was not adequately highlighted in the current manuscript) is to complement the limited understanding of flood characteristics at the global scale (e.g. changes in flood magnitude and timing). Specifically, recent observation-based findings on flood hazard magnitude and timing (Do et al., 2019; Do et al., 2017; Hodgkins et al., 2017; Burn and Whitfield, 2018; Mangini et al., 2018; Hall and Blöschl, 2018; Slater and Villarini, 2016) have not provided much information for Asia (note that China contributes a significant share to Asia land area). The key barrier is the insufficient streamflow records presented with the most comprehensive global datasets to-date such as GRDC or GSIM (Do et al., 2018a; Gudmundsson et al., 2018a), and thus large-scale hydrologists do not possess a representative sample of streamflow observation (especially for Asia). Current understanding of global flood hazards may therefore be biased toward Europe and North America, making studies similar to the present study very demanding.*

**Response:** We thank the reviewer for this excellent suggestion. We have substantially revised the manuscript, and highlight the missing piece of research on flood hazards over China in the Introduction part (Line 32-36 in the revised manuscript). We clearly describe in the Introduction that "*Due to the limitation of observational datasets, existing knowledge on flood hazards is significantly biased towards Europe and North America, with the characteristics of other worldwide regions far from being well represented*" by following the reviewer's suggestion. Suggested references have been included where necessary. We further emphasize the contribution of our analysis based on this unprecedented dataset to global-scale flood hydrology at the very end of the manuscript. Thanks!

*(5) If the authors prefer to stay on the current objective (i.e. conducting an analogous study of a previous study), I want to see this objective introduced more prominent (e.g. explain why a nearly identical study was conducted) as it is currently mentioned very vaguely in the introduction (page 2, lines 35-37). This potentially makes a notion that the authors did not give Villarini and Smith (2010) the merits it deserves.*

**Response:** The manuscript has been pre-reviewed by Gabriele Villarini and James Smith before submission. Both of them are long-term collaborators of the leading author for the present study. We extend our thanks to their comments in the Acknowledgement section.

The objective has been revised. The strategy is to target questions that are unique and important to flood hazards over China (see also response to comment #3), even though some of the approaches have been used in Villarini and Smith (2010) and other previous studies. Thanks!

*(6) The authors should also focus on clarifying the technical aspects that were identically repeated and the ones that were done differently (beside different datasets and study areas) – the rationale of these choices should also be discussed. In addition, "distributions" in Villarini and Smith (2010) refer to several terminologies (e.g. spatial, temporal and statistical distributions) and the rationale of using this word was presented adequately in the introduction of that paper. The authors must rethink the key motivation of their study to clarify which "distributions" are the focus of their research and justify why a specific characteristic of flood peaks is chosen (similar to the link between Typhoon Nina and timing of flood peaks currently presented). The loose ties between the "distributions" and the objectives currently made the paper read fragmented and confused.*

**Response:** This is a very good suggestion, and has been adopted in developing the revised manuscript. We clarify the aspects of flood peak distributions that we are planning to focus on at the very beginning of the manuscript. We also note that understandings of these aspects would contribute to improved characterization of flood hazards over China from both statistical and physical perspectives. An important difference of our study from Villarini and Smith (2010) is analysis on the impacts of landfalling tropical cyclones on the upper tails of flood peaks. This is mainly motivated by that facts that some of the most extreme floods in the history of China are associated with tropical cyclones (e.g., Typhoon Nina) and also the unique nature of China's geographic location on the margin of the most active oceans in Tropical cyclones. These issues have been clarified or emphasized in the Introduction section. Thanks!

*(7) If the authors decide to follow another approach, the study may exclude some analyses (e.g. GEV distribution) to focus more on an aspect of flood hazards (e.g. trends in flood magnitude and timing) and relate the paper findings to previous observation-based studies. As indicated above, with this unprecedented dataset, I believe there are several rooms for analyses and discussions beyond the current strategy (i.e. an analogous study of Villarini and Smith (2010)). There are opportunities for deeper and more critical discussions included in the "Specific comments" section that are applicable if the authors choose this direction.*

**Response:** We thank the reviewer for this critique. The objective has been revised, but still demonstrate the focus of flood hazards. We believe that the upper tail of flood peaks based on GEV framework is an important characteristic of flood hazards, as it represents how extreme floods are statistically distributed. Equally important is the impact of different flood agents on the upper-tail properties. We therefore prefer to keep the analyses in the revised manuscript. The strategy is not to present an analogous study of Villarini and Smith (2010). Please see comments #3, #5, and #6. Thanks!

All the specific comments raised by the reviewer have been properly incorporated in the revised manuscript. Thanks!

*(8) Abstract: This section should be revised significantly to provide a clear synthesis of the objectives, the methods being used and the main findings. The opening sentence reiterates the ambiguous terminology presented in the title (i.e. "flood peak distributions") without sufficient clarifications in the rest of the abstract. There are several statistical analyses presented with little "hints" about the role of each assessment as well as the links between the results and specific objectives. This shortcoming has led to the notion that the paper is a collection of unrelated analyses (which is not the case when reading further to the main text).*

**_Response_**_:_ We substantially reconstruct the Abstract by following the reviewer's comments. Thanks!

_(9) Introduction: After reading through the introduction twice, I'm still unclear about "what are the research questions being introduced?", and "what are the key contributions of this study?" The confusions partly come from the loose link between the ideas being presented and the choice of terminologies/wordings that sometimes overcomplicates the ideas. This section must be revised to provide a more synthesized literature review and simplify the ideas where possible. Please also refer to my previous comments for some thoughts about improving the introduction. The authors may find some of the "specific comments" useful as well._

**_Response_**_:_ We substantially reconstruct the Introduction section by following the reviewer's comments. The current structure starts with the main objective of this study and a brief introduction of key aspects of flood peak distributions to be examined. We move on to the following paragraphs through literature review that shows existing gap in our knowledge in flood hydrology and hydroclimatology. We conclude the section by raising five questions to be targeted at for the following analysis (Line 88-93). The questions echo with Summary and Conclusions. Thanks!

_(10) Results and Discussions: This section was too "report focused" with very loose connection between individual sub-sections. This led to a notion that the "Results and Discussion" is a compilation of four separate studies. I strongly recommend the authors to remedy this issue and better explain the contribution of each sub-section to the overarching aims. One possible way is to have an "introduction sentence/paragraph" at the beginning of each sub-section to clarify how the subsequent discussion contributes to a better understanding of "flood peak distributions for China"._

**_Response_**_:_ We thank the reviewer for this critique. We remove the content about changes in flood peak timing, since we decide to just focus on flood peak magnitudes (including long-term changes, seasonality, upper tails etc.) in the present study. The logistic structure of the remain contents is explicitly described at the very beginning of Results and Discussion section (Line 166-173 in the revised manuscript). We also place one or two sentences at the beginning of each subsection to hint on the relationships between the present subsection and previous subsections. Thanks!

_(11) Conclusions: This section reads more "Summary" than "Conclusion". It would be more useful if the authors be more concise in summarizing the key findings and then focus on highlighting (1) what was the contribution of this study to the state-of-understanding for flood hazard (China-wide and global scale) and (2) the fact that collected data set is unprecedented and has much more potentials beyond this study (e.g. adding values to the literature of global scale hydrology). It would beneficial to also comments on the benefit, challenges and potential of making China streamflow data becomes more FAIR (Findable, Accessible, Interoperable and Reusable; see Wilkinson et al., 2016). Some potential options toward this ambitious goal is to publish metadata and indices (Do et al., 2018b; Gudmundsson et al., 2018b) and even include hydrological variable time series (Addor et al., 2017)._

**_Response_**_:_ We thank the reviewer for this critique. We substantially reconstructed the Conclusion section in the revised manuscript. We first change the title of this section to "Summary and Conclusions". We highlight the main contribution of our present study to improved understanding on flood hazards across China (e.g.,

long-term changes, upper tails and physical drivers), and emphasize innovative approaches for future flood frequency analysis that can explicitly address the nonstationarities in flood series across China. The role of tropical cyclones in flood hydroclimatology contributes to physical insights into the upper tails of flood peaks across China. We conclude this section (and our paper) by highlighting the further utilization of this exceptional dataset in flood hydrology from a global perspective by following the reviewer's suggestion. Thanks!

*(12) Methodological issues There are some data aspects that should be clarified in Section 2 (or presented in supplementary) as this is the first study to use such dataset, which may be unfamiliar to other hydrologists.*

**Response***:* We expand this section by providing more details about the dataset in the revised manuscript. Thanks!

*(13) Data quality statements. The definition of "continuous records of at least 50 years" (e.g. non-missing data consecutively for 50 years) and "Strict quality control procedures are implemented to ensure consistency and accuracy of the records". These aspects are very important on the credibility of detectable trends.*

**Response***:* We emphasize this in the revised manuscript by following the reviewer's suggestion. Thanks!

*(14) Filtering criteria related to missing data. Please note that annual value of maximum is very sensitive to missing data. Previous studies usually applied a threshold of number of missing data points per year to assign "N/A" value for a specific year (e.g. station A has 30 missing data points in 1950 so N/A value was assigned for that particular year). If all gauges have zero missing data-point, this feature should be highlighted to show the exceptionally good quality of this dataset.*

**Response***:* We thank the reviewer for this critique, and totally agree with this argument. However, we note that the flood records from China demonstrate miscellaneous ways of data collection, rather than based on instantaneous discharge with regular temporal intervals (like most U.S. geological survey stations). However, since the stations are all national control-stations with the highest data quality, various procedures are implemented to make sure each of the annual statistics are reliable and accurate. For some stations, annual maximum is the only available data. We expand the text on quality control in the revised manuscript (Line 99-106). Thanks!

*(15) A graph presents the number of stations with available data over time may be useful. This will also provide an understanding of the uncertainty related to step-change analysis. Figure 2 of Do et al. (2017) is an example for this type of plot.*

**Response***:* We add a sub-plot in Figure 2 by following the reviewer's suggestion. Thanks!

*(16) Figure 1 should show the location of "discarded stations" prior to further analyses. It seems to me that the station density reduced significantly in other figures. If all stations showed in Figure 1 were used, please note in the caption. If there are stations removed, the number of stations used/removed for each subsequent*

*analysis should also be mentioned.*

**Response:** Figure 1 shows all the 1120 gauges used throughout the manuscript. We clarify this in the caption. Only stations with results being statistically significant are shown in subsequent figures, which leads to reduced numbers of stations compared to Figure 1. We clarify this in the caption of each subsequent figure. Thanks!

*(17) Additional maps should be added to Figure 1 to show (i) data length, (ii) the beginning and (iii) end of records for each station to complement the subsequent analysis of change-point. For instance, one hypothesized reason for change point detected mostly over 1980-2010 was most stations have streamflow data available for only 1970s onward.*

**Response:** We show record length of each station in Figure 1 using shaded color. We note that approximately 95% stations show end of records after 2016, we therefore prefer not to show this in the map. The beginning of records can also be inferred based on record lengths or Figure 2a (the time series of total number of available stations for each year). Approximately 90% stations have records during the period 1960-2017 instead of from 1970s onward. Thanks!

*(18) Trends analysis: as a China-wide investigation is one of the key motivations of this study, I'm recommending the use of "reference period". The authors may conduct the analysis for only one period (e.g. all stations were assessed for 1969-2019) or different period (e.g. 1980-2010, 1950- 2019, and 1900-2019). This is particular important to remedy data limitation, which could dismiss the usefulness of the detected trends wherever too short records was used to assess trend in floods over a long period (e.g. using 50 years of data to infer changes in floods for a 150-year period).*

**Response:** The trend analysis is based on time series with record lengths exceeding 50 years which should be quite enough to infer robust slowly varying changes in flood peaks. In addition, more than 95% stations extend onward after 2016, with about 90% stations covering the entire period 1960-2017. We believe additional analysis focusing on "reference period" is therefore not necessary. Thanks all the same!

*(19) For Mann-Kendall test: is there any criterion for data length to conduct the analysis (e.g. at least ten data points)? This is relevant for the test conducted for before-step-changes and after-step-changes time series (there may be insufficient data points). I noted that the figures 5b and 5c have different numbers of stations, potentially due to insufficient data points for a specific sub-group?*

**Response:** The reviewer is absolutely correct. Only sub-groups with record lengths exceeding 10 years are used for further trend analysis. We add this information in the revised manuscript (Lines 135). Different numbers of stations in Figure 5b and 5c are partially due to the fact that some of the sub-series do not show statistically significant trends before change points, in contrast to the sub-series after change points. Thanks!

*(20) For flood timing, it is unclear the motivation for assessing trends in this flood index. In addition, it is more useful to use an analysis where the magnitude of changes is visible. The reason is even when changes are significant, it is not practically meaningful if flood timing shifts only a small time-window (e.g. one day*

*over 100 years). Theil-Sen slope estimator is a useful statistical technique for this type of analysis (Blöschl et al., 2017).*

**_Response_:** The reviewer is absolutely correct. We decide to remove all the materials pertaining to the trends in flood peak timing from the revised manuscript. We are working on a parallel manuscript that specifically focus on the seasonality of annual flood peaks across China. Thanks!

*(21) Title: "Flood Peak Distributions over China" made the notion that "spatial distribution" is the key feature being analyzed (which is not the case).*

**_Response_:** We respectfully disagree with the reviewer on this particular point. We use the word "distributions" similar to Villarini and Smith (2010) by referring to a couple of meanings that include spatial and temporal distribution, seasonal distribution, and statistical distribution. This is also pointed by the reviewer under comment #8. We clarify its meanings in the Abstract and Introduction sections. Thanks all the same!

*(22) Line 2: "flood peak distributions across China" in the abstract also made the notion that "spatial distribution" is the key feature being analyzed.*

**_Response_:** We clarify the multiple meanings of "distributions" in the Abstract section. Thanks all the same!

*(23) Line 16: the research objectives were "to provide improved understandings on the nature of upper tails of flood peaks and innovative methods for flood frequency analysis in a changing environment". This statement has two issues: (1) it is unclear what is "the nature of upper tails of flood peaks"? (Any alternative for "upper tails of flood peaks", which was used quite often across the manuscript?), and (2) "provide innovative methods for flood frequency analysis" indicates the development of a new method, which is not the focus of this paper. The goal should be rewritten.*

**_Response_:** Description on the research objectives has been removed. We note in the revised manuscript that "The ultimate goal of our study is to provide improved characterizations of flood hazard across China from both statistical and physical perspectives". Thanks!

*(24) Line 17: It is unclear how the four presented themes linked to the two objectives.*

**_Response_:** The four themes have been removed in the revised manuscript. Thanks!

*(25) Line 26: "staionarity" should be "stationarity".*

**_Response_:** Done. Thanks!

*(26) Line 35: "highlight possible factors that induce the changes" should be clarified: some of potential factors and why did they are chosen?*

***Response****:* We change it to "highlight potential influencing factors" in the revised manuscript. Thanks!

*(27) Line 36: it is unclear to me what "dominant modes of violation for the stationarity assumption" is. Would there be a simpler way to explain it?*

***Response****:* We actually mean "whether abrupt changes or slowly varying trend is the main demonstration of nonstationarities in flood series". We add "abrupt changes and slowly varying trend" into the context of this sentence, but still prefer to use "dominant mode" for simplicity. The terminology is also used in previous studies (Villarini and Smith, 2010). Thanks!

*(28) Line 37: the final sentence seems out of place. I was expecting a clarification of the "dominant modes..." mentioned it the previous sentence, or justification of why the authors followed Villarini and Smith (2010) rather than Hodgkins et al. (2019).*

***Response****:* This sentence has been moved to an earlier place. Due to unavailability of meta-data (including changes in land use/land cover, regulation for each drainage basin), we are unable to attribute the changes in flood series to specific factors as Hodgkins et al. (2019). We are working on developing a meta-data archive for all the gauges, and expect to do analysis as Hodgkins et al. (2019) in future studies. Thanks!

*(29) Line 46: it is unclear what "space-time rainfall organizations" means.*

***Response****:* We replace it with "spatio-temporal rainfall variability" in the revised manuscript. Thanks!

*(30) Line 48: I don't think the sub-sequent sections discussed anything related about "the necessity of improved procedures for regional flood frequency analysis with spatial heterogeneity in flood hydroclimatology considered". Please revise this statement or extend the discussion.*

***Response****:* This sentence has been removed in the revised manuscript. Thanks!

*(31) Lines 52: please note that "seasonality" may refer to more than the timing of floods (Villarini, 2016).*

***Response****:* We remove "(e.g., seasonality)" in the revised manuscript. Thanks!

*(32) Lines 57-62: please focus more on introducing the mechanisms generating floods across China.*

***Response****:* This sentence has been removed in the revised manuscript. We focus on the flood-generation mechanisms in China by following the reviewer's suggestion. Thanks!

*(33) Line 63: it is unclear what "monsoon-related systems" means.*

***Response****:* We changed it to "monsoon" in the revised manuscript. Thanks!

*(34) Lines 64-67: these two sentences read out of place.*

**Response*:* These sentences have been removed in the revised manuscript. Thanks!

*(35) Line 67-68: this section is repetitive. I'm recommending the authors to link this paragraph to the subsequent one (starting at line 70) as timing of tropical cyclones were then related to flood timing.*

**Response*:* These sentences have been removed in the revised manuscript. Thanks!

*(36) Lines 70-92: I found the detail introduction of Typhoon Nina is too disruptive-and a distraction. This is also a reason for confusion about the key research question of this research. If the key motivation was "to examine the impact of tropical cyclone on the upper tail properties of flood peak distribution over China" (line 85; and I think this is a great research question itself), the structure of the manuscript need to be revised to reflect this main research objective. The current manuscript presents the link between floods and tropical storms at the very end (section 4.4), after a very long discussion for other factors (e.g. step-changes, seasonality, GEV...) with almost no "reminder" for readers. At the time I got to section 4.4, I almost forgot the motivation to analyze "Tropical cyclones and upper tail properties" in this study.*

**Response*:* We shrink the discussion on Typhoon Nina in the revised manuscript. We substantially reconstruct the Introduction section by summarizing the key research questions we are targeting in the present study. The structure of the Results and Discussion sections has also been reconstructed by connecting each piece of analysis in a better way. Thanks!

*(37) Line 88: I'm not sure why "Results presented in this study can promote a predictive understanding of flood hazards associated with landfalling tropical cyclones"*

**Response*:* This sentence has been removed in the revised manuscript. Thanks!

*(38) Line 121: please explain the rationale of applying Mann-Kendall test for two sub-groups and the contribution of this analysis to the main objective.*

**Response*:* We note that monotonic trends can be induced by abrupt changes in mean rather than indicting slowly varying trend in the series. We therefore do trend analysis for series without significant change points in mean. For those series with change points in mean, we do trend analysis for two sub-series. MK test for the sub-groups can highlight stations that demonstrate both abrupt changes in mean and slowly varying trend in the entire flood series. We reconstruct this paragraph pertaining to trend analysis in the revised manuscript (Line 128). Thanks!

*(39) Line 124: "significance level of 5%" is for one-tail or two-tail?*

**Response*:* It is two-tailed test that we are using throughout the manuscript. We make this clear in the revised manuscript. Thanks!

*(40) Line 125: please justify why circular statistic, a common approach for flood timing investigations (Villarini, 2016; Blöschl et al., 2017; Hall and Blöschl, 2018), was not used in this study to prevent the issues where two flood peaks occurring on calendar day 1 and calendar 365 only have one day difference.*

**Response**: We throw out the analysis on the changes in flood peak timing in the revised manuscript. We are currently working on an ongoing manuscript that specifically focus on the seasonality of annual flood peaks across China, including changes in flood peak timing. Thanks!

*(41) Lines 136-140: this is repetitive (has been mentioned in introduction). The authors should clarify about how to examine the dependence of GEV parameters on drainage area.*

**Response**: We reconstruct this section by deleting the repetitive texts. The dependence is examined based on checking the correlations of three GEV parameters with drainage areas as shown in Figure 9. Thanks!

*(42) Line 145: is there any reference for this choice (i.e. 500km and two weeks).*

**Response**: We add several references in the revised manuscript (Line 155). Thanks!

*(43) Lines 149-155: these information seems irrelevant as only the circulation center location was used. In case the sub-sequent analyses will divide tropical storms into sub-categories (ET/TS), the authors should clarify this technical aspect.*

**Response**: One of the distinctive features of tropical cyclones that produced a large number of flood peaks over China is that they experience extratropical transition during their life cycles. We provide an explicit discussion about this feature in section 4.4 (also see Table 1 for details). We prefer to keep this information in the manuscript. Thanks all the same!

*(44) Line 160: "The majority of stations tend to show smaller values". Please clarify what is "the majority" and how "smaller" the values are (perhaps in %).*

**Response**: There are 398 and 305 stations that exhibit smaller values of mean and variance after than before the change point, respectively. We have clarified this in the revised manuscript (Line 178). We prefer not to provide specific values in the manuscript, as they show considerable variation by stations. Thanks!

*(45) Line 169: "We are able to relate some of the changes in annual flood peaks series to intentional human activities". Please clarify the procedure used to identify these relationships (e.g. metadata inspection?) and what is the magnitude of "some" (e.g. 5% of stations with significant step-change detected?).*

**Response**: The reviewer is correct. The abrupt changes in flood series can be related to intentional human activities based on meta-data analysis. We clarify this in the revised manuscript (Line 187). Due to the limitation of meta-data, we are only able to relate the changes for the four stations shown in Figure 4. We believe these four representative stations demonstrate some of the main facets of human regulations in the flood series, although additional analysis is needed. This calls for additional efforts in data collection and

interpretation, which seems quite challenging at present. Thanks!

*(46) Figure 3: Please also show the locations of gauges with insignificant results. The geographical location of individual stations analyzed in Figure 4 should also be highlighted (e.g. starred symbol).*

**Response:** We thank the reviewer for this suggestion. However, we prefer not to show all the locations, as by doing this the figure will turn a complete mess. We show all the stations in Figure 1 for readers' reference. The difference between Figure 3 and Figure 1 will be stations that show insignificant results. We highlight the locations of the four stations analyzed in Figure 3 using numbers in brackets. Thanks!

*(47) Line 197: this is somewhat expected as changes in climate variables occur quite gradually in general. Previous studies generally link abrupt changes to human interventions rather than natural climate drivers.*

**Response:** This is not entirely true. One of our previous studies found abrupt changes in rainfall series across China (Yang et al., 2013; similarly, also see Gu et al., 2017a). Although we are not quite sure of the drivers (i.e., shift in climate), changes in rainfall can obviously demonstrate themselves in flood series. However, the spatial patterns between changes in rainfall and changes in flood series are not consistent, which lead us to conclude that climate may play a less important role in flood series across China. We do not modify the text. Thanks all the same!

*(48) Line 203: please be careful with your conclusion that "Abrupt change rather than slowly varying trend is a common mode of the violation of the stationarity assumption for the annual flood peak series over China" as both modes of non-stationarity can present at the same station. For instance, "naturalized streamflow records" (e.g. the difference in means between two before/after step-change time series were removed) may reveals gradual change, which is more relevant to climate changes and variabilities. As a result, assessing linear trends over only stations that did not exhibit significant step-change is not sufficient to support this statement.*

**Response:** We thank the reviewer for this critique. Note that we are comparing "abrupt change" vs. "**slowly varying** trend", rather than linear trends. Linear trends can be induced by both abrupt change points in mean (as the example of naturalized streamflow series raised by the reviewer) or indicate slowly varying trend for a time series. That is the reason why we detect abrupt change points, and then move on to slowly varying trend detection for those series without significant abrupt changes in mean. We believe our results can support our argument that "abrupt change rather than slowly varying trend is a common mode of the violation of the stationarity assumption across China". We clarify this in the methodology section of the revised manuscript. Thanks!

*(49) Figure 5b and Figure 5c: please clarify why these two figures have different number of data points. If it is due to statistical insignificance, please also show the location of stations that did not exhibit significant linear trends.*

**Response:** Different numbers of data points between Figure 5b and 5c can be resulted from (1) insufficient record length (less than 10 years) for the sub-series before or after change points, (2) linear trends being not

statistically significant. The figures would turn into a 'chaos' after we show all stations. We therefore prefer to only highlight data points that are statistically significant. We add explanations for the difference in the number of stations in the caption of Figure 5. Thanks!

*(50) Line 227: Please note that the findings do not support the statement that "external climate factors (i.e., extreme rainfall), and changes in soil moisture on flood hydrology" leads to flood stationarity.*

**Response:** We rephrase this sentence, and now it reads "Attribution analysis on the nonstationarities of annual flood peaks across China point to mixed controls of human activities, external climate factors (i.e., extreme rainfall), and changes in soil moisture on flood hydrology". The argument is partially based on comparative analysis between the present study and previous studies that focus on changes in extreme rainfall across China (i.e., Yang et al., 2013, Gu et al., 2017a; 2017b). Thanks!

*(51) Lines 232-235: it is unclear what "state-of-art process-based approaches" and "statistical modelling approaches" are – please clarify. Please also make it clearer why these approaches important to "flood frequency analyses across China" (i.e. how could these approaches address the nonstationarities in flood frequency).*

**Response:** We reconstruct this section to more clearly explain what process-based and statistical modelling approaches really refer to. These innovative approaches can explicitly deal with nonstationarity in flood series which is the case for most stations across China, especially northern China (refer to Figure 3 and Figure 5b). Thanks!

*(52) Section 4.2: maps of the average flood timing and associated concentration (Villarini, 2016) would be a nice addition.*

**Response:** We thank the reviewer for this suggestion. We decide to reserve more details about the seasonality of flood peaks across China to an ongoing study. Thanks!

*(53) Lines 239: "the first peaks"? Considering the distribution of the floods timing (Figure 6), I thought there is only one peak per group?*

**Response:** It should be "the first peak". We correct this in the revised manuscript. Thanks!

*(54) Lines 240-241: please provide reference.*

**Response:** Done. Thanks!

*(55) Line 252: please clarify how "tropical cyclone floods" defined (this should be presented in methodology).*

**Response:** We have clarified how to associate an annual flood peak with tropical cyclones in the manuscript (i.e., section 3.3). We remind readers by pointing them to the methodology section in the revised manuscript.

Thanks!

*(56) Line 260: forward-reference is not recommended.*

**Response:** We removed these sentences in the revised manuscript. Thanks!

*(57) Lines 265-280: without sufficient evidence of "how many days flood peaks have shifted", it is hard to justify these statements.*

**Response:** This section has been completely removed in the revised manuscript. Thanks!

*(58) Line 271: "Villarini (2016) found ..." sounds out of place.*

**Response:** We removed this sentence in the revised manuscript. Thanks!

*(59) Lines 273-278: these statements read contradicting to each other.*

**Response:** These sentences have been removed. Thanks!

*(60) Line 287: please clarify why only these stations used (should explain in the methodology).*

**Response:** The requirement of probability theory is the data samples should be identically and independently distributed. We thus only conduct GEV analysis for stations that are show stationary flood records during the study period. We clarify this in methodology section by following the reviewer's advice (Line 146-147). Thanks!

*(61) Line 312: "contrasting space-time organizations" is unclear. Please clarify.*

**Response:** We replace it with "Spatial contrasts in rainfall climatology between northern and southern China" in the revised manuscript. Thanks!

*(62) Line 343: the presented results only show the impacts of tropical cyclone on flood occurrence rather than "flood peak distributions".*

**Response:** We respectively disagree with the reviewer on this particular point. Note that we highlight stations with records floods (scatter) together with the frequency of annual flood peaks (shade) induced by tropical cyclones in Figure 10. Record flood is the largest flood peak for the entire record, and points to the upper tail of flood peak distribution. The argument is further supported by testing the dependence of GEV shape parameter on tropical cyclones. We do not modify the text. Thanks all the same!

*(63) Figure 12: please plot also the stations within 500-km distant and the annual maximum streamflow does not coincide with the occurrence of the selected tropical cyclones (i.e. flood timing is outside the two-week*

*threshold) and extend the discussions appropriately (e.g. the proportions of stations influenced by tropical storms).*

**Response**: We thank the reviewer for this suggestion. However, we prefer not to show all the stations with annual flood peaks not associated with tropical cyclones. We provide a figure below to show that the add-in will make annual-flood-peaks stations less standing-out (Figure R2a). We instead highlight the 500 km buffer zone for each tropical cyclone in the revised manuscript (as the one showed in Figure R2b). We add the number of total storm-affected stations in Table 1, and expand the discussion in the revised manuscript. Thanks!

[Figure]

Figure R2. Sample figure to compare visual effects of two ways of dealing with other stations: (a) include all other stations, (b) add a buffer zone.

***References***

[revised manuscript text omitted]

---

## Author Comment (AC3) · 16 Sep 2019

**Responses to Reviewer 2**

*(1) The authors present an extensive study on exploring flood peak distributions in terms of stationarity, seasonality, scaling property, spatial heterogeneity and the effects of landfalling tropical cyclones across the China. Their results are mainly based on several statistical methods (e.g. Pettitt's test, Mann-Kendall test, GEV distribution, etc.) using annual maximum peak discharge from over 1000 gages with a record of at least 50 years. To my knowledge, their study is the first to analyze the characteristics of annual maximum flood peaks on a nation-wide scale in China, highlighting the importance of both streamflow dataset and treating them not only as numerical values but as real world physical events. Overall, this study is organized and well-written. However, I do have some major concerns, which are listed below, about the structures and certain elements of the study that need to be addressed before the work can be considered for publication. Typographical errors (e.g. line 255, "Tainhang Mountains" should be "Taihang Mountains") in the paper can be addressed in the second round of review because some of them might be removed during author's revision.*

**Response:** We are really grateful to the reviewer's appreciation on our manuscript. We have revised the manuscript based on all the comments from both reviewers. We apologize for the typos, and take a thorough check on the revised manuscript. Thanks!

*(2) The main objective or scientific question of this study are not adequately addressed that causes each part of analysis does not connect logically. Authors have applied several statistical approaches with peak flood data but the explanation of the necessity and connection of each test need to be more explicitly addressed. I understand that similar study was done by Villarini and Smith (2010) US but the authors need to demonstrate why Villarini and Smith's analysis are also necessary in this study. For example, why do authors decide to explore the role of tropical cyclone rather than snowmelt in characterizing the upper tail of flood peaks?*

**Response:** We thank the reviewer for this critique. We substantially reconstruct Abstract and Introduction sections to highlight the objectives and scientific issues of the present study in the revised manuscript. In short, we expect to provide improved characterizations of flood hazard over China from both statistical and physical perspectives, and contribute to improved understandings of flood hydrology and hydroclimatology under a changing environment. Our analysis is based on an unprecedented dataset that can significantly advance flood science at the global scale. Thanks!

Most of the analyses implemented in Villarini and Smith (2010) are also frequently used in previous studies. Some of the methods are the norm for this type of analysis. For instance, examining upper tails of flood peaks based on the GEV distribution. As can be shown from Figure 6, snowmelt contributes little to the population of annual flood peaks which is a striking contrast to the seasonality of flood peaks in the US where we see a large portion of annual flood peaks is due to snowmelt (Smith et al., 2018). Thanks all the same!

*(3) Although authors conclude that abrupt changes in flood magnitude and seasonality are mainly due to anthropogenic influence, I would suggest them to investigate (or even focus on) climate induced*

*nonstationarity. The anthropogenic induced changes imply stationarity after changing point (e.g. the built of reservoir, urbanization), which is not valid in this study, as shown in Figure 5, highlighting the existence of climate driven changes. The author's explanation for Figure 5c is vague, such as these negative trend might be due to soil conservation or decreasing rainfall intensity. That is why I suggest authors to separate climate-induced changes from anthropogenic-induced one when discussing the violation of stationarity in China. The other reason is that climate-induced changes around the world has already been unearthed, and author's contribution might provide such understandings in China. For example, Blöschl et al. (2017, 2019) found changing climate derives changes in flood seasonality and magnitude over Europe using streamflow data. Blöschl, Günter, et al. "Changing climate shifts timing of European floods." Science 357.6351 (2017): 588-590. Blöschl, Günter, et al. " Changing climate both increases and decreases European river floods." Nature 1476. 4687 (2019): https://doi.org/10.1038/s41586-019-1495-6.*

**Response:** We thank the reviewer for this critique. As we have stated in the introduction, the present study does not aim to distinguish the signals of climate-driven and anthropogenic-driven that lead to nonstationarities in flood series across China. Instead, we expect to highlight all potential factors that lead to the changes. Separation climate-induced changes from anthropogenic-induced changes involves collective efforts in modeling and extra datasets, e.g., Berghuijs et al. (2019), and is a current study underway. We reconstruct the Introduction section to emphasize the objectives of the present study. Thanks!

The decreasing trends in flood series as shown in Figure 5c can be possibly be due to soil-conservation practices. This is based on one of previous studies by Bai et al. (2016) that focus on changes in runoff series. Again, we highlight it as a possible reason, but also mention additional investigation is needed. The excellent studies by Blöschl et al. (2017, 2019) might be also interesting to transfer to China, but we sense that climate might play a secondary role in changing flood regimes in China. Thanks!

*(4) A minor relevant suggestion is to modify the label in Figure 7b, indicating clearly which trend represents flood seasonality shifting to earlier or later time in a year.*

**Response:** The figure has been removed for being not logistically connected to other content in this study. Thanks!

*(5) Despite the significance of flood seasonality as authors addressed, the section 4.2 of this paper and other relevant parts failed to present how flood seasonality is distributed across the country and how to link them to flood generating mechanisms. I would suggest authors to add two maps: average seasonality of all maximum annual floods and of only the three (or two) biggest floods for all gages across the country. These two maps can indicate the regional patterns of flood seasonality and how flood processes may change as one moves from moderate to extreme floods. They can also help authors demonstrate some of their arguments, such as line 240-242, "Frequent occurrence of annual" Without a map, it is very difficult for reader to envision how flood seasonality is distributed across the country.*

**Response:** We thank the reviewer for this critique. We decide to particularly focus on the role of tropical cyclones in determining contrasting upper tails of flood peak distributions across China. An ongoing study is focusing seasonality of annual flood peaks based on Circular statistics. We greatly shrink the text on flood

peak timing by removing analysis pertaining to the trends. However, we still provide texts on the mixture of flood-generation mechanisms across China, but mainly pointing readers to classic textbooks rather than present details. Thanks!

*(6) An interesting finding in this paper is that tropical cyclones (monsoon-controlled storms) plays key role in determining upper tail of flood peaks in northern (southern) China (line 252-264), but needs to be well defended. Figure 6 only shows that flooding happens more frequently around June (July) in southern (norther) China, but fails to present their severity (magnitude). For instance, is it possible the most frequent floods in June over southern China associate with low or moderate severity (i.e. magnitude)?*

**Response**: We thank the reviewer for this critique. We reconstruct this section by following the suggestion of Reviewer 1 (mainly to avoid forward-reference). The contrasting determining factors of upper tails of flood peaks between northern and southern China are well illustrated by Figure 10 and Figure 11 where we see tropical cyclones produce most record floods in northern China (Figure 10) with the shape parameters of GEV distribution regulated by the presence of tropical cyclones (Figure 11).

The author is correct! The shaded color in Figure 10 shows that there are more frequent flood peaks induced by tropical cyclones in southern China, but they do not represent upper tails (i.e., less severe than flood peaks induced by monsoon). Thanks!

*(7) Line 54-56 ("Annual flood peaks resulted. . ."): I suggest being more careful in your wording here. I would respectfully disagree with authors that conventional flood frequency analysis (FFA) requires a homogenous flood population with respect to flood generating mechanisms. Instead, a flood series at any gage is a mixture of different flood-generating processes but they are just one sample not the population. Conventional FFA assumes peak discharge measured at a gage over a finite period is a sample from the population of all possible floods (representing different flood generating mechanisms) during an undefinable length of time.*

**Response**: This sentence has been removed in the revised manuscript. Thanks!

*(8) The authors should provide a brief description of the quality control procedures in section two rather than just state they have done so. This procedure is important in this study since datasets are from different sources and inaccessible to public. For instance, what is the time interval for the "instantaneous" peak discharge and are they the same value for all gages? If the instantaneous peak discharge data were from US Geological Survey (USGS), I would expect they are all in the 15-minute interval but have no clue here.*

**Response**: We thank the reviewer for this critique. Stream gauging stations from China demonstrate a variety of ways of data collection rather than based on extraction from instantaneous discharge at regular temporal intervals (like most USGS stations). For some of the stations, the annual maximum flood peak is the only available data. We have expanded the text on data quality control in the revised manuscript (Line 99-106). Thanks!

**References:**

Berghuijs, W. R., Harrigan, S., Molnar, P., Slater, L. J., & Kirchner, J. W. The relative importance of different flood-generating mechanisms across Europe. Water Resources Research, 55, 4582–4593, 2019

[revised manuscript text omitted]

---

## Author Response (AR2)

Dear Dr. Xing Yuan,

Re: Manuscript #HESS-2019-322, "On the Flood Peak Distributions over China" by Yang et al.

We would like to express our sincere thanks to Hong Do for detailed comments and suggestions which substantially help improve the manuscript. We provide a point-by-point response to their comments below. The reviewer's comments are enumerated. Our replies to each comment start with "***Response***". We look forward to hearing from you at your earliest convenience.

Yours sincerely,
Long Yang

Address: School of Geography and Ocean Science, Nanjing University, Nanjing 210023, Jiangsu Province, China.
Email: yanglong86123@hotmail.com

**Responses to Reviewer 1**

*(1) In the revision, the authors have improved the narrative and structure of the manuscript to provide a stronger link across all analyses. The additional materials (e.g. number of gauges influenced by storms in Table 1) also provide clarification for many aspects of the analyses. The decision to remove flood timing analysis particularly made the study more streamlined and better highlight the key contributions. Before publication, I have some further comments to be addressed. From my perspective, these comments are fairly straightforward and thus a minor revision is recommended.*

**Response***:* We really appreciate the reviewer's efforts and time on our manuscript. We revise the manuscript substantially based on all the comments, and make a point-by-point response below. The reviewer's comments are enumerated. Our replies to each comment start with "**Response**". Thanks!

*(2) Data description: the extended description from line 101 to 106 reads quite vague. I'm particularly confused about the statement "For instance, the dates of annual maximum flood peak and highest stage should be comparable, with records of missing flood peak timing discarded to ensure data accuracy". Does this mean that the authors (or data providers) compared the annual maximum date identified from (i) discharge and (ii) stage records – and for years that either one of them is not available the data points are removed? The author should revise this statement.*

**Response***:* We have reconstructed this paragraph in the revised manuscript. We decide not to mention stage record, as obviously it is not relevant in our analysis. Thanks!

*(3) A good way to make data quality aspect more transparent is to provide references for data quality control procedure (e.g. a national standard for hydromet information) or include a supplementary table of the original sources (or data providers) for each subset of the whole database. Some information that could be included is the contact (or website) of data providers, data access policy (freely accessible or there is a payable fee) and the practices that were used for quality control (where possible).*

**Response***:* Flood records are through quality-control procedures by following the code for hydrologic data compilation of China (SL247-1999). We add this in the revised manuscript. Thanks!

*(4) Section 4.2 now reads alienated from the whole manuscript, mostly due to the removal of most materials related to flood timing analysis. To remedy this issue, a potential way is to add a short paragraph at the data section, mentioning that record of both the magnitude and timing of flood peaks was used for this assessment, of which the timing was used to highlight the mixture of flood-generation mechanisms.*

**Response***:* We add one sentence ("*In addition to flood peak magnitude, flood peak timing (i.e., date of occurrence for flood peaks) is also provided, and is mainly used to infer flood-generating mechanisms over different regions across China*") in the manuscript. Thanks!

*(5) In addition, I am not sure how Figure 6 could be interpreted as "There are three (two) distinct peaks in the seasonal distribution of annual flood peaks for southern (northern) China" as the three lines essentially show only one prominent peak (between date number 150 and 200 for southern China and between 200 and 250 for northern China and caused by TCs). The authors should revise the description to better communicate the key feature of Figure 6. In addition, the x-axis should also be revised to be consistent with the description (i.e. use Jan-Dec rather than 1-365).*

**Response:** We redesign Figure 6 in the revised manuscript, with the peaks of flood frequency highlighted in the figure. Thanks!

*(6) Figure 3 caption should include the percentage of stations showing significant trends (currently mentioned in the main-text only) to ensure it can be used as a stand-alone element. If prospective readers want to have a brief overview of "how floods have changed across China", this figure and the improved caption should be the sufficient answer.*

**Response:** Done. Thanks!

*(7) Figure 5 caption should include the criterion for "insufficient record lengths" (e.g. less than 5 data points?).*

**Response:** We add "(e.g., less than 10 years)" in the caption of Figure 5. Thanks!

*(8) Figure 7 caption should include the criterion for "stationary stations" (i.e. insignificant change points in mean or in variance or monotonic trends).*

**Response:** Done. Thanks!

*(9) The final recommendation (about the future data archive) should be extended to specify the future development of the database (e.g. develop a website to make data available or implement the practice of the published hydrological databases).*

**Response:** Instead of specifying future efforts, we cite Do et al. (2018) which provides a good example of making hydrological data publically accessible. Thanks!

[revised manuscript text omitted]